# Bacteria evolve macroscopic multicellularity by the genetic assimilation of phenotypically plastic cell clustering

Yashraj Chavhan [1] ✉, Sutirth Dey[2] & Peter A. Lind [1,3] ✉

The evolutionary transition from unicellularity to multicellularity was a key innovation in the history of life. Experimental evolution is an important tool to study the formation of undifferentiated cellular clusters, the likely first step of this transition. Although multicellularity first evolved in bacteria, previous experimental evolution research has primarily used eukaryotes. Moreover, it focuses on mutationally driven (and not environmentally induced) phenotypes. Here we show that both Gram-negative and Gram-positive bacteria exhibit phenotypically plastic (i.e., environmentally induced) cell clustering. Under high salinity, they form elongated clusters of ~ 2 cm. However, under habitual salinity, the clusters disintegrate and grow planktonically. We used experimental evolution with *Escherichia coli* to show that such clustering can be assimilated genetically: the evolved bacteria inherently grow as macroscopic multicellular clusters, even without environmental induction. Highly parallel mutations in genes linked to cell wall assembly formed the genomic basis of assimilated multicellularity. While the wildtype also showed cell shape plasticity across high versus low salinity, it was either assimilated or reversed after evolution. Interestingly, a single mutation could genetically assimilate multicellularity by modulating plasticity at multiple levels of organization. Taken together, we show that phenotypic plasticity can prime bacteria for evolving undifferentiated macroscopic multicellularity.

The evolutionary shift from unicellular organisms to multicellular ones represents an important gateway towards innovation in the history of life[1]. This shift has conventionally been categorized as a 'major evolutionary transition' because it created a new level of biological organization that natural selection could act on[2], which likely facilitated an unprecedented increase in biological complexity[3,4]. Here we focus on the evolution of the capacity to form undifferentiated cellular clusters, which was likely the first key step towards the evolution of multicellularity and has evolved independently in at least 25 distinct lineages across the tree of life[1,3,5]. Discerning the nuances of this evolutionary transition is inherently difficult because it occurred in deep past >2 billion years ago. Most transitional forms have likely

undergone extinction, and the scarcity of fossil evidence severely limits what can be gleaned about this transition. In the face of severely limited fossil evidence, experimental evolution has proven to be a very powerful tool in this regard as it can combine empirical rigor with diverse experimental designs to directly observe the unfolding of this transition in action[6–14]. However, the existing experimental evolution studies of multicellularity have two major limitations.

First, although multicellularity first evolved in prokaryotes[15], most studies that experimentally evolve multicellularity in unicellular ancestors have dealt with eukaryotes (e.g., unicellular fungi[6–9], algae[10–13], and more recently ichthyosporeans[14]). Even though multicellularity has evolved independently in three distinct bacterial lineages in the history

[1]Department of Molecular Biology, Umeå University, Umeå, Sweden. [2]Indian Institute of Science Education and Research (IISER) Pune, Pune, India. [3]Umeå Centre for Microbial Research, Umeå University, Umeå, Sweden. ✉e-mail: yashraj.chavhan@umu.se; peter.lind@umu.se

of life[3,16], the experimental evolution studies of de novo prokaryotic multicellularity have been largely restricted to mat formation in *Pseudomonads*[17]. In contrast to the de novo evolution of prokaryotic multicellularity, a rich body of work has investigated the nuances of the *already* evolved prokaryotic multicellularity[18–20]. Some particularly striking examples include the fruiting bodies of myxobacteria[21], filamentous growth with cellular differentiation in cyanobacteria[22], and the complex hyphal networks of streptomycetes[23]. Thus, the scarcity of prokaryotic experimental evolution studies represents a key gap in the current understanding of the evolution of multicellularity, which we aim to address here.

Second, most experimental evolution studies on multicellularity focus on mutationally derived (not environmentally induced) multicellular phenotypes[6,9,12,14,24]. These studies have revealed that the mutations required to form undifferentiated multicellular clusters are relatively easily accessible in diverse unicellular eukaryotic taxa. Moreover, a wide variety of environmental conditions can selectively enrich such de novo mutations (e.g., predation[10,11,13], diffusible stressful agents[25], improved extracellular metabolism[6], etc. (reviewed in ref. 5)). Interestingly, novel phenotypes like multicellular clusters can also be expressed in the absence of mutations when such phenotypes are induced by environmental changes.

Phenotypic plasticity, which enables a given genotype to express different phenotypes in different environments[26–28], can be an important source of evolutionary novelty throughout the tree of life[29–31]. This is because plasticity can facilitate biological innovation by allowing genes to be 'followers' in the evolution of new phenotypes[32,33]. Specifically, plastic phenotypes can be genetically assimilated when selection enriches mutations that make their expression constitutively expressed, even in the absence of environmental induction[34–36]. Thus, phenotypic plasticity has the potential to accelerate evolutionary innovation by bypassing the wait for mutations required for new beneficial phenotypes. Phenotypic plasticity can also play an important role in the evolution of multicellularity. For example, environmentally induced stress responses have been co-opted to evolve the germline-soma differentiation in the multicellular alga *Volvox carteri*[37,38]. Similarly, the environmentally induced cAMP-based stress response has been co-opted for multicellular development in the social ameba *Dictyostelium discoideum*[39]. While the co-option of unicellular ancestral traits/pathways has been an important theme in the evolution of multicellularity[40–42], many such unicellular traits were phenotypically plastic, and their expression has evolved from an environmentally induced (temporal) to developmental (spatial) context[38]. Importantly, diverse taxa exhibit facultative (phenotypically plastic) multicellular phenotypes. For example, phytoplankton[43], cyanobacteria[44], and *Pseudomonads*[45,46] can facultatively form multicellular clusters in response to predation. Moreover, a recent study has shown that changes in environmental salinity can induce multicellular clustering in marine cyanobacteria[47]. Given such widespread facultative multicellular clustering, plasticity has the potential to directly facilitate the evolution of multicellularity. This can happen if the ancestrally facultative cell clustering can be assimilated (i.e., the development of clusters is made environmentally robust and insensitive). Indeed, such assimilation (without its exact genetic basis) has been demonstrated in the unicellular alga *Chlamydomonas reinhardtii*[11] (also see ref. 13), which facultatively forms microscopic clusters comprising -140 cells in the presence of rotifer predators. However, no other experimental evolution study has conclusively demonstrated that phenotypic plasticity can facilitate the evolution of multicellularity. Two specific questions remain unanswered in this context: (1) Can phenotypic plasticity facilitate the evolution of *macroscopic* multicellularity (comprising large clusters with >10⁴ cells)? (2) Can it do so in bacteria? Our study addresses both these questions empirically.

Here we show that both Gram-negative and Gram-positive bacteria exhibit environmentally induced macroscopic cell clustering.

Using experimental evolution in both shaken and resting conditions, we artificially selected for macroscopic clusters and disfavored planktonic growth while progressively reducing the environmental induction for clustering. Our experiment shows that phenotypic plasticity can facilitate the evolution of macroscopic multicellularity in bacteria by bypassing and avoiding the wait for mutational emergence of undifferentiated cluster formation. We demonstrate that phenotypically plastic cell clustering in ancestral genotypes can be rapidly assimilated to efficiently form multicellular clusters even in the absence of the environmental induction. We also elucidate that phenotypically plastic clustering is also manifested at the level individual cell shapes. Finally, we show that mutations in a small number of genes linked to the cell wall can genetically assimilate the ancestral phenotypic plasticity at multiple levels of organizations, ultimately leading to obligately multicellular bacterial life histories.

## Results

### Both Gram-negative and Gram-positive bacteria exhibit phenotypically plastic cell clustering

We observed that high salinity liquid environments can make both Gram-negative and Gram-positive bacteria grow primarily as elongated macroscopic clusters and not as turbid cultures of individual planktonic cells (Fig. 1). Specifically, we grew independent clonal cultures of *Escherichia coli* (Gram-negative) and *Staphylococcus aureus* (Gram-positive) in two distinct environments (Luria Bertani (LB) broth containing either 0.5% or 6% NaCl (w/vol)) in unshaken tubes at 37 °C (see *Methods*). Henceforth, we refer to these two environments as "habitual salinity" and "high salinity", respectively. These two bacterial species have putatively diverged from their common ancestor >3000 million years ago. As expected, under habitual salinity, both *E. coli* and *S. aureus* showed planktonic turbid growth without any observable clustering (Fig. 1; Supplementary movies 1 and 2). In contrast, under high salinity, both *E. coli* and *S. aureus* grew predominantly as elongated clusters and not as planktonic cultures (Fig. 1; Supplementary Movies 3 and 4). In both species, the clusters comprised >10⁵ viable colony forming units (CFUs) and reached 2−3 cm in length when cultured in tubes containing 5 ml nutrient medium. Such clustering was phenotypically plastic (environmentally induced): when transferred to a habitual salinity environment, the clusters disintegrated into individual cells that grew planktonically (Supplementary Fig. S1). High-resolution time-lapse videos of macroscopic cluster formation revealed that in static high salinity environments, both *E. coli* and *S. aureus* showed a combination of clonal and aggregative modes of multicellular growth (Supplementary movies 3 and 4). Put differently, the multicellular growth under high salinity was a consequence of bacterial cells staying together after division (clonal expansion) *and*

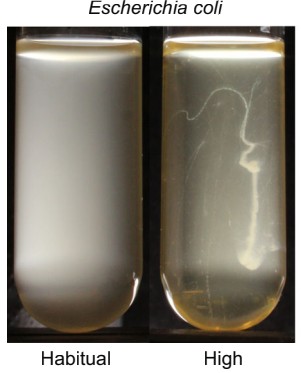

*Escherichia coli*                    *Staphylococcus aureus*

Habitual    High            Habitual    High
Salinity                    Salinity

**Fig. 1** | Both *E. coli* (Gram-negative) and *S. aureus* (Gram-positive) show the capacity to form phenotypically plastic elongated macroscopic cell clusters.

previously unattached cells (or cellular clusters) adhering to each other (aggregative growth).

We further established that both *Citrobacter freundii* and *Pseudomonas aeruginosa* also exhibit such environmentally induced cell clustering, suggesting that it is widespread in bacteria (Supplementary Fig. S2). However, the formation of elongated clusters is not a physically inevitable outcome of bacterial growth under high salinity: the Gram-negative bacterium *Serratia marcescens* did not exhibit such phenotypic plasticity and grew as a turbid planktonic culture under both habitual and high salinity (Supplementary Fig. S2). This led us to investigate if the phenotypically plastic bacterial clustering was itself an evolvable biological phenomenon.

Since the emergence of undifferentiated clusters is expected to be the first key step towards the evolution of multicellularity[1,3,5], we hypothesized that phenotypically plastic cell clustering could facilitate the evolution of undifferentiated multicellularity in bacteria. Focusing on *Escherichia coli*, we set out to determine if the clustering induced by high salinity can be genetically assimilated into an obligately multicellular bacterial life history, even in the absence of environmental induction.

### Experimental evolution of simple multicellularity via the genetic assimilation of phenotypically plastic cell clustering

We established that *E. coli* could form phenotypically plastic macroscopic clusters not only in resting tubes but also in well-mixed environments where the culture tubes were shaken at ~180 rpm (Supplementary Fig. S3). We used a single *E. coli* MG1655 colony to propagate two distinct experimental evolution lines (S (for S̲haken) and R (for R̲esting)) to artificially select for increased macroscopic cell clustering in environments with progressively reducing salinity (Fig. 2a; see *Methods*). Specifically, for both S and R lines, we picked a small portion of the previous day's bacterial cluster fitting within a 20 μl aliquot and washed it four successive times in sterile 2 ml media before transferring it to a fresh culture tube. This diluted the planktonic bacteria by at least $10^8$-fold while keeping the macroscopically clustered bacteria undiluted. Thus, macroscopic clustering was favorable in *both* shaken and resting conditions in our artificial selection scheme. Propagating five replicate populations per line, we started the evolution experiment with media containing 6% NaCl (w/vol) and progressively reduced the salt concentration over 50 days (see *Methods*).

The rationale behind conducting experimental evolution in both resting and shaken conditions is that these two environments can select for qualitatively different clustering. This is because in resting cultures, oxygen supply depletes steeply from the air-liquid interface to tube's floor. Hence, selection for increased clustering in resting cultures is likely to enrich mutants that cluster preferentially at the air-liquid interface[48]. In contrast, such oxygen availability gradients are much weaker in shaken tubes, where selection for greater clustering may not enrich interface inhabiting mutants.

Unlike most other evolution experiments, here the phenotype of interest (macroscopic cluster formation) was already exhibited by the ancestor at the outset (induced by high salinity). We hypothesized that artificial selection for clustering under progressively reduced salinity should enrich mutations that can make the clustering relatively less dependent on environmental induction. This expectation mirrors the "genes as followers" view of phenotypic evolution[33]. At the end of the evolution experiment, we tested if clones from the evolved populations were able to make macroscopic clusters in static habitual salinity environments (i.e., without environmental induction; see *Methods*).

Our evolution experiment resulted in the successful genetic assimilation of the ancestrally plastic phenotype in most of the evolved S and R lines (Fig. 2b; Supplementary movies 5 and 6). Specifically, clones representing 4 out of five S lines (S1, S2, S4, and S5) and 4 out of five R lines (R1, R2, R3, and R5) grew as macroscopic clusters even in

the absence of environmental induction (Fig. 2b; Supplementary movies 5 and 6). Moreover, all five replicates of both S and R retained their ancestral ability to form elongated clusters in high salinity environments (Fig. 2b; Supplementary movies 7 and 8). Furthermore, under both habitual and high salinity, a large majority of viable cells belonging to the evolved lines were found within clusters and not within the planktonic phase (Supplementary Fig. S5). Interestingly, the macroscopic clusters formed under habitual salinity were not elongated: S1, S2, S4, and S5 made a large number of macroscopic clusters that sank upon rapidly growing in size (Supplementary movie 5). The habitual salinity environment offers a weaker buoyant force than the high salinity environment; this could explain why the macroscopic clusters formed by S1, S2, S4, and S5 under habitual salinity were not elongated like the clusters formed by these clones under high salinity. In contrast to the S clones, R1, R2, R3, and R5 each formed a single mat (~1 mm thick) at the air-liquid interface under habitual salinity (Supplementary Fig. S4; Supplementary movie 6). Moreover, the R1, R2, and R5 mats remained intact throughout the growth phase (Supplementary Fig. S4); these mats disintegrated and sank only upon external perturbation, as shown in Fig. 2b. Thus, selection for clustering without environmental induction in resting tubes indeed enriched mutants that preferentially grew at the air-liquid interface, as we had hypothesized initially.

Since S1, S2, S4, S5, R1, R2, R3, and R5 formed multicellular clusters even in the absence of environmental induction, we conclude that they successfully evolved the first step towards multicellularity which demands that cells *inherently* grow as clusters.

Our selection protocol made the bacterial clusters undergo an artificially imposed life cycle where a small piece of the cluster in question (which was disintegrated by vigorous vertexing and then transferred into fresh media) gave rise to a new (larger) cluster. This motivated us to test if the clusters also qualify as biological units that could *spontaneously* complete a life cycle consisting at least one multicellular stage[49–51]. To this end, we first cultured an evolved S clone (S2) in an arena where the bacteria could access fresh nutrients without being artificially transferred using a pipette. We found that the bacteria successfully completed a life cycle where the old clusters spontaneously gave rise to new clusters after accessing fresh nutrients (Supplementary movie 9). Using a different arena, we further demonstrated that another S clone (S5) can successfully complete such a such life cycle under both habitual and high salinity (Supplementary movie 10).

Taken together, the genetic assimilation of ancestrally plastic cell clustering led to the evolution of undifferentiated multicellularity in our experiments, which enabled bacteria to grow inherently as multicellular units, even in the absence of environmental induction. Having investigated phenotypic plasticity and its genetic assimilation at the level of *collectives* of cells (clusters), we turned our attention to the effects of selection on phenotypes at the level of *individual* cells.

### Phenotypic plasticity and its evolution at the cellular level

We performed both brightfield and fluorescence microscopy on the ancestral and evolved clones to determine if and how macroscopic cluster formation corresponded to changes in the cell shape (see *Methods*). We found that *E. coli* shows stark phenotypic plasticity in cell shape between habitual and high salinity environments (Fig. 3). Specifically, whereas the ancestral genotype showed its characteristic rod shape under habitual salinity, its cells became spherical under high salinity (Fig. 3). Surprisingly, we found that all the evolved lines lost their spherical cell shape under high salinity and their cells became elongated (Fig. 3). We quantitatively analyzed these cellular morphological changes using two distinct metrics (see *Methods*).

The ancestral genotype showed significant phenotypic plasticity in terms of the cellular perimeter observed in 2d images: specifically, the ancestor had significantly smaller cells under high salinity than

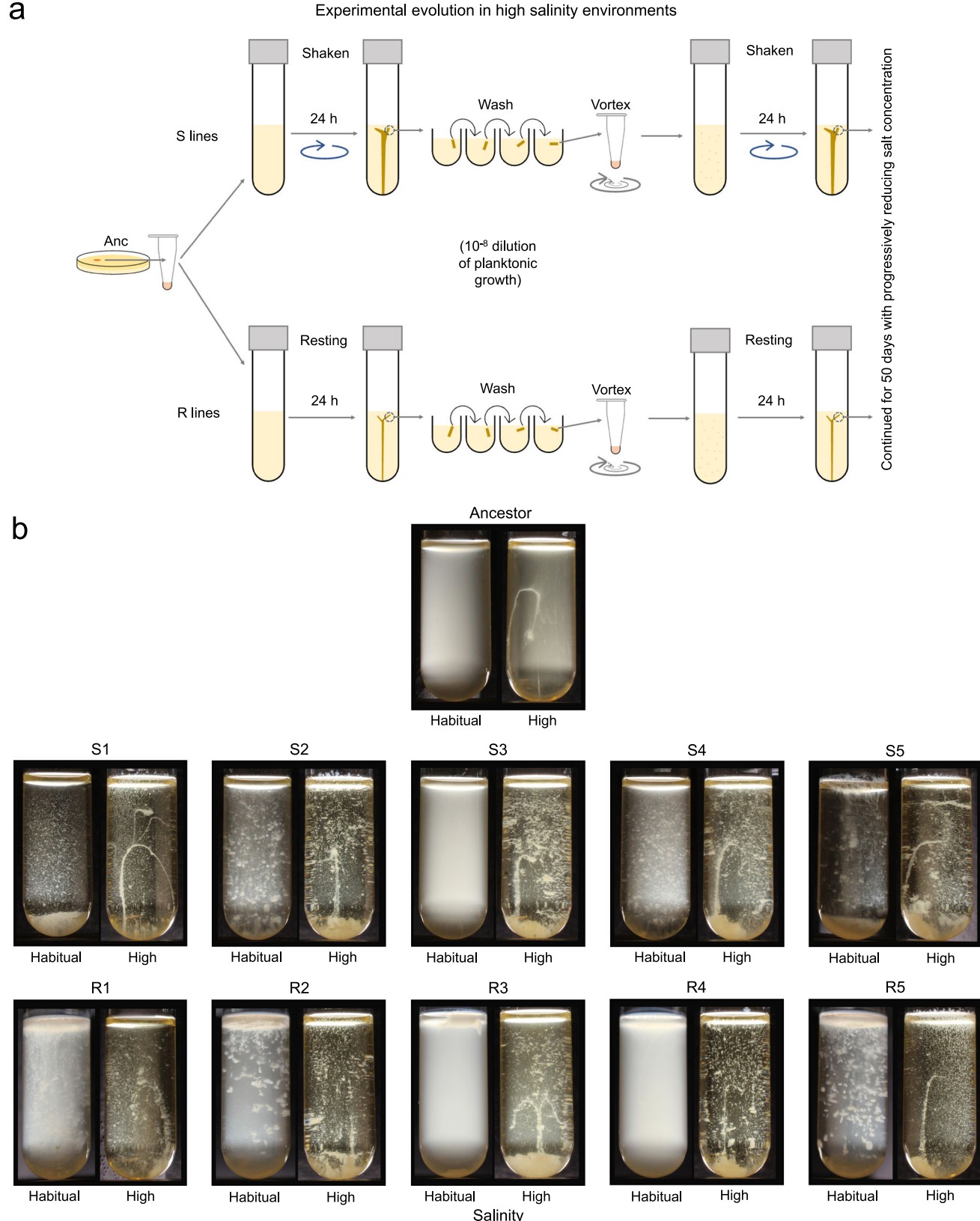

**Fig. 2 | Experimental evolution of macroscopic multicellularity. a** A schematic of our experimental evolution workflow. **b** Clonal phenotypes at the end of experimental evolution after growth under static conditions. Also see Supplementary movies 1 & 3 (for Anc), 5 & 7 (for the S clones), and 6 & 8 (for the R clones). In R1-R5, the habitual salinity tubes were externally perturbed at the end of the growth cycle to disrupt mats formed at the air-liquid interface and show cell clustering (see Supplementary Fig. S4 for the unperturbed tubes).

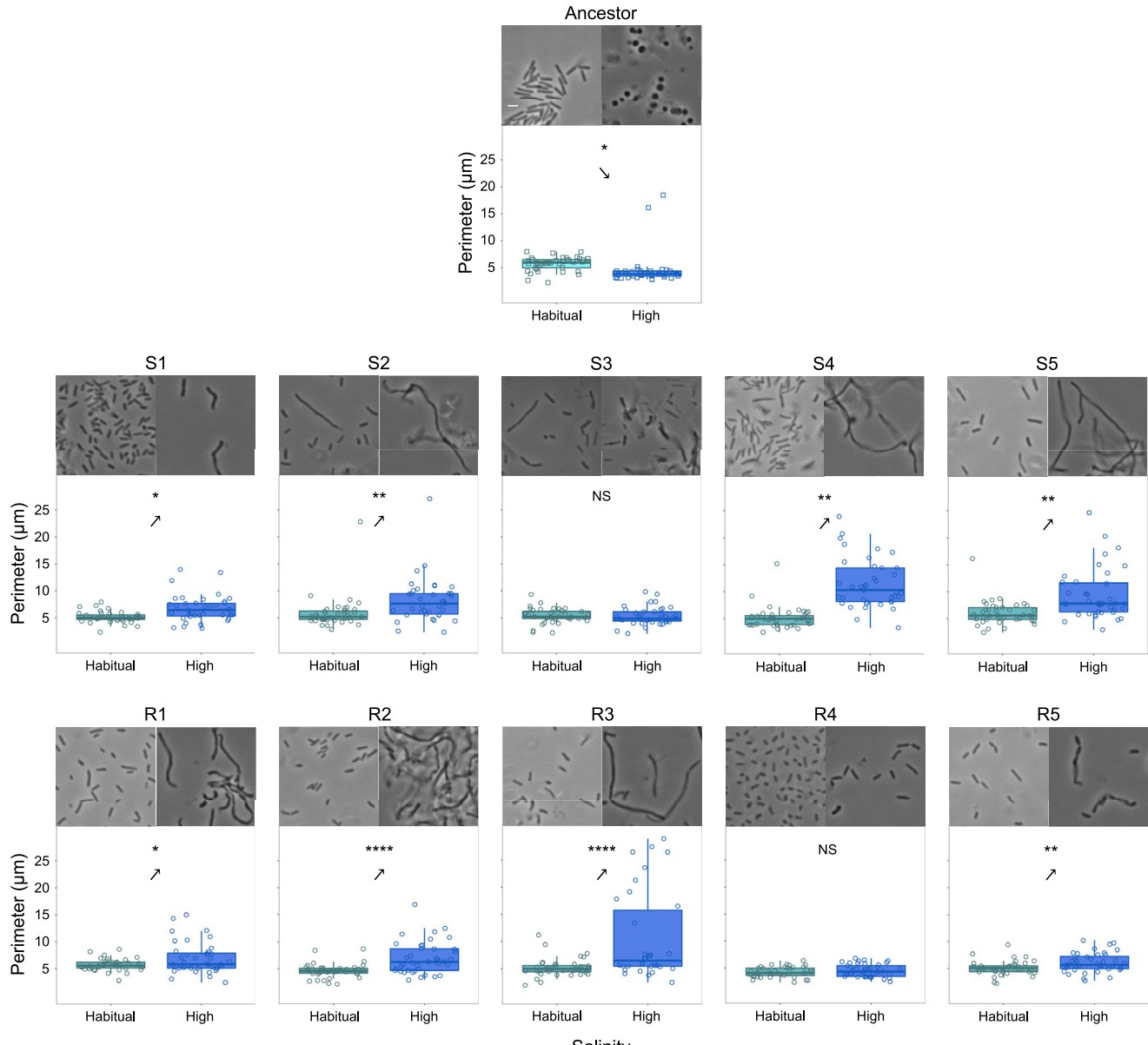

**Fig. 3 | The evolution of phenotypic plasticity in cellular morphology.** The arrows point towards the qualitative direction of phenotypic plasticity. The lower and upper box hinges show the 25 and 75% quantiles, respectively; the thick horizontal band represents the median. The lower whisker denotes the smallest observation ≥ lower hinge − 1.5 × interquartile range; the upper whisker represents the largest observation ≤ upper hinge + 1.5 × interquartile range. Two-tailed *t*-tests (unequal variance across types): *$P \leq 0.05$; **$P \leq 0.01$; ***$P \leq 0.001$; ****$P \leq 0.0001$.

See Supplementary Table S1 for statistical details (exact *P* values). The reversal of the ancestral cell perimeter plasticity (observed in eight out of 10 evolved clones) corresponded to the genetic assimilation of phenotypically plastic cell clustering (compare with Fig. 2b). All the plotted data are provided in the Source Data file. Also see Supplementary Fig. S6 for cell shape plasticity quantified in terms of cellular circularity.

under habitual salinity (Fig. 3; Supplementary Table S1). In contrast, clones representing 4 out of five S lines (S1, S2, S4, and S5) and 4 out of five R lines (R1, R2, R3, and R5) showed a reversal of the ancestral phenotypic plasticity in terms of the cell perimeter (Fig. 3; Supplementary Table S1). Specifically, S1, S2, S4, S5, R1, R2, R3, and R5 showed significantly larger cells under high salinity (Fig. 3; Supplementary Table S1). We found a clear correspondence between reversal of the cell perimeter plasticity and successful genetic assimilation of cellular clustering: The eight lines that showed a *reversal* in the ancestral cell perimeter plasticity were also the ones that successfully genetically assimilated the cellular clustering during experimental evolution (compare Figs. 2 and 3). On the other hand, the remaining two lines (S3 and R4) which showed no significant difference in cell

perimeters under habitual versus high salinity were also the ones that failed to successfully assimilate macroscopic clustering (compare Figs. 2 and 3).

We also analyzed cellular morphology in terms of the circularity of individual cells (see *Methods*). The ancestor showed significant phenotypic plasticity in terms of cell circularity: rod shaped cells under habitual salinity versus spherical cells under high salinity (Supplementary Fig. S6; Supplementary Table S2). In contrast, the cells belonging to S1, S3, R4, and R5 underwent moderate elongation that imparted the characteristic rod shape of *E. coli* under both habitual and high salinity (Supplementary Fig. S6). We also found that 5/10 lines (S1, S3, R1, R4, and R5) exhibited similar cell circularity across habitual and high salinities (Supplementary Fig. S6; Supplementary Table S2).

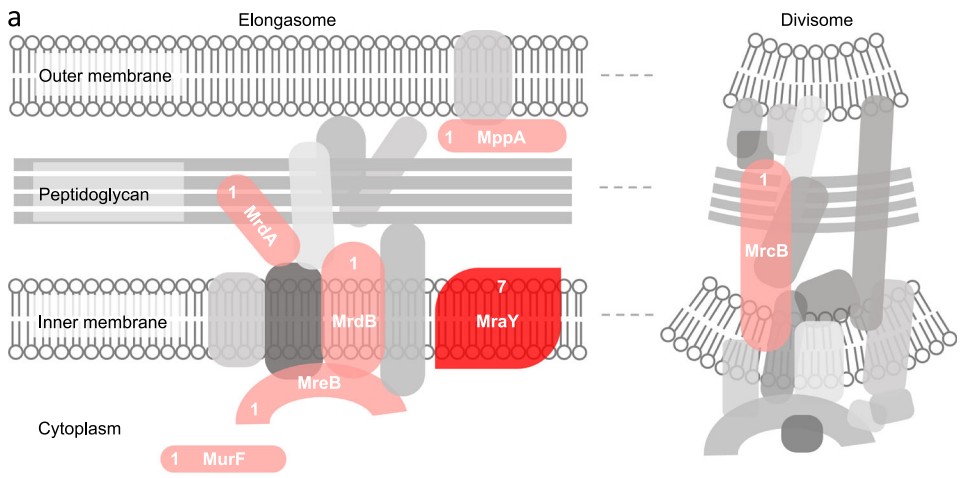

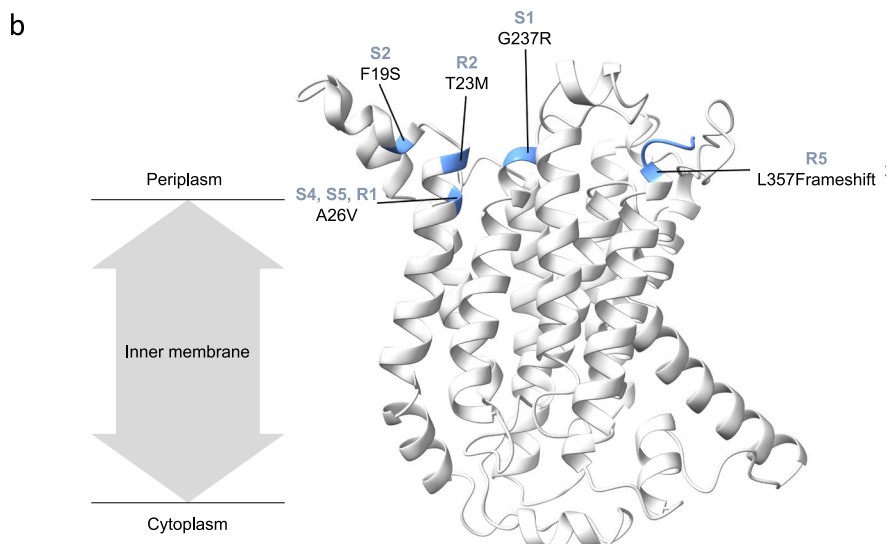

**Fig. 4 | The genetic basis of assimilated multicellularity. a** Experimental evolution of genetically assimilated multicellularity primarily enriched mutations in genes involved in cell wall assembly. The schematic shows the proteins encoded by the mutated genes in red. The numbers accompanying the mutated proteins represent the number of clones that showed a mutation in a particular protein. Two genes (*murF* and *mppA*) showed synonymous mutations. **b** The location of mutations on the 3D structure of MraY, the protein that mutated in 70% of the sequenced clones. All the mutated regions are located near the periplasmic region of the transmembrane protein.

Moreover, a relatively greater cellular elongation in the other 5 lines (S2, S4, S5, R2, and R3) under high salinity reversed their ancestral phenotypic plasticity in terms of cellular circularity and made them filamentous (Supplementary Fig. S6; Supplementary Table S2).

We further found that plastic phenotypes evolved in opposite directions at the level of cell collectives and individual cells (Supplementary Fig. S7). Specifically, at the level of individual cells, the evolved cell shape under both habitual and high salinity for both S and R matched the *uninduced* ancestral cell shape (i.e., the evolved cells were non-spherical; Supplementary Fig. S7). In contrast, at the level of cell collectives, both S and R exhibited cell clustering under both habitual and high salinity (Fig. 2 and Supplementary Fig. S5). Moreover, in both S and R, the fraction of viable cells found within multicellular clusters was relatively closer to the *induced* ancestral value (Supplementary Fig. S7). Thus, at the level of cell collectives, the evolved (and genetically assimilated) phenotype matched the *induced* ancestral phenotype (multicellular clustering; Supplementary Fig. S7). Taken together, we showed that plastic phenotypes can evolve in opposite directions at different levels of biological organization.

Having investigated phenotypic plasticity and its genetic assimilation at two distinct levels of biological organization (multicellular units vs. individual cells), we studied the genetic basis of the evolution of undifferentiated multicellularity observed in our experiments.

### The genetic basis of assimilated multicellularity in *E. coli*

We sequenced whole genomes of all the S and R clones described in Figs. 2 and 3 and compared them to the ancestor to identify the mutations that resulted in the evolution of simple macroscopic multicellularity in our experiments (see *Methods*). We found that most mutations occurred within genes involved in the biosynthesis of peptidoglycan, which forms the bulk of eubacterial cell walls (Fig. 4a). Specifically, out of the 19 mutations putatively linked to changes in cell surface properties, 13 were found within genes directly involved in peptidoglycan biosynthesis (Supplementary Data 1). We found that 7 out of the ten sequenced clones had a mutation in MraY, the enzyme that catalyzes the first membrane-bound step of peptidoglycan biosynthesis[52]. Despite such high degree of parallelism at the level of genes, we found several different mutations at widely distributed

locations within the primary chain of MraY (Supplementary Data 1). Interestingly, all these mutations were concentrated towards one side of the tertiary structure of MraY, facing the periplasmic zone of the transmembrane protein (Fig. 4b). These mutations, spaced apart from the cytoplasmic active site of MraY by the bacterial inner membrane, likely play a role in recruiting other peptidoglycan-related proteins in the periplasm. We found that all the seven clones with an MraY mutation successfully evolved simple macroscopic multicellularity by genetically assimilating the ancestrally plastic cellular clustering (compare Figs. 2b and 4b). In addition to an MraY mutation, 6 of these seven clones also carried mutations in other genes linked to peptidoglycan synthesis, biofilm formation, or adaptation to nutrient media (Supplementary Data 1). Moreover, we observed the strongest genetic assimilation of the clustering phenotype (with almost all bacterial growth within macroscopic clusters and the absence of detectable turbidity) in the S clones that carried at least one mutation in addition to an MraY mutation (S1, S2, and S5; see Fig. 2b and Supplementary Data 1).

Curiously, S4 was only one MraY mutation away from the common ancestor (Supplementary Data 1), which suggests that a single mutation can be sufficient for genetically assimilating the ancestrally plastic cell clustering. It is worth noting that such assimilation driven by a single mutation was relatively weak: the S4 clone showed a combination of macroscopic clusters and planktonic growth under habitual salinity (Fig. 2b). Furthermore, R3, the only clone that evolved a multicellular life history without enriching an MraY mutation, also displayed a weak genetic assimilation characterized by a combination of both clustering and turbid growth (Fig. 2b).

Unlike the S clones, the R clones preferentially colonized the air-liquid interface under habitual salinity (Supplementary movie 6). We found that the mutations in the R clones could potentially explain this phenotypic difference. Specifically, R5 had two mutations in genes putatively linked to mat formation at the air-liquid interface through c-di-GMP signaling (*dgcQ* and *pdeA* (Supplementary Data 1)). Moreover, R2, R3, and R4 had mutations within (or upstream to) genes with possible links to biofilm formation (*mprA* encoding a transcriptional repressor (R2, R3, and R4) and *bhsA* encoding an outer membrane protein (R4); Supplementary Data 1). Furthermore, none of the S clones showed a mutation in any of these four genes linked to interface inhabiting mat formation (*dgcQ*, *pdeA*, *mprA*, or *bhsA*). Such mutational contrast could potentially explain the differences in the abilities of the S and R clones to inhabit the air-liquid interface.

Several other genes linked to peptidoglycan biosynthesis which mutated in our study (*mrcB*, *mrdA*, *mrdB*, *mreB*, *murF*; Fig. 4a) are linked to the maintenance of cell shape in *E. coli*[53–55]. Specifically, both MrdA and MrdB are known to play key roles in maintaining the characteristic rod shape of *E. coli*[53,54]. Moreover, MreB, which is the bacterial analog of actin, is an essential protein that forms a scaffold which interacts with several other peptidoglycan biosynthesis proteins and plays key role in cellular elongation[52]. Finally, MraY has been shown to affect both the cell shape and adhesion in the multicellular cyanobacterium *Anabaena*[56]. This suggests that the mutations observed in the clones that successfully assimilated multicellular clustering can also be linked to the evolutionary changes in cell shape plasticity (Fig. 3; Supplementary Data 1).

### The biochemical basis of plastic clustering and its evolution

To determine the biochemical nature of the molecules that bind bacteria within clusters in our study, we assayed the ability of two distinct hydrolytic enzymes[57] (cellulase and proteinase K) to inhibit cell clustering, under both habitual and high salinity (see *Methods*).

In the ancestral genotype, cell clustering (which occurs only under high salinity) could be successfully reduced by cellulase but not by proteinase K (Supplementary Fig. S8; Supplementary movie 11). Although cellulase could not completely inhibit macroscopic clustering,

it discernably reduced the formation of elongated filament-like clusters and increased the turbidity of the ancestral broth (Supplementary Fig. S8; Supplementary movie 11). Thus, β−1,4-glycosidic linkages (but not peptide linkages) played a key role in binding the ancestral cells within clusters under high salinity.

Surprisingly, in the evolved clones, we found that the bonds keeping the cells together within clusters were qualitatively different in the presence and absence of environmental induction. Under high salinity, macroscopic clustering in the evolved clones was successfully reduced by cellulase but not by proteinase K (Supplementary Figs. S9 and 10; Supplementary movie 12). Cellulase prevented the formation of elongated filament-like clusters in the evolved clones under high salinity, but it could not completely inhibit macroscopic clustering (Supplementary Figs. S9 and S10; Supplementary movie 12). Hence, similar to the ancestor, clustering in the evolved clones under high salinity was also mediated by β−1,4-glycosidic linkages, but not by peptide linkages.

In contrast to the above, under habitual salinity, proteinase K completely inhibited macroscopic clustering in 7/8 genotypes with genetically assimilated clustering (Supplementary Figs. S9 and S10; Supplementary movie 13). Moreover, while cellulase could not completely inhibit uninduced macroscopic clustering, it resulted in markedly reduced clusters at the air-liquid interface in 7/8 genotypes with genetically assimilated macroscopic clustering (Supplementary Figs. S9 and S10; Supplementary movie 13). Thus, clustering in the evolved clones under habitual salinity was mediated by both peptide and β−1,4-glycosidic linkages.

Taken together, the biochemical basis of the genetically assimilated (uninduced) clustering was distinct from that of induced clustering. Thus, although macroscopic clustering was genetically assimilated as a phenotype at the level of cell collectives, the underlying lower-level phenotype (i.e., the nature of bonds keeping cells together without environmental induction) was distinct from that of the original induced phenotype. This shows that the genetic assimilation of a higher-level phenotype can be brought about by the expression of contrasting lower-level phenotypes.

## Discussion

Our study begins with the demonstration that bacteria show phenotypically plastic cell clustering that results in large macroscopic structures in high salinity environments. Since both Gram-negative and Gram-positive bacteria exhibit this phenomenon (Fig. 1), such plastic development of multicellular clusters appears to be a common (but not universal) bacterial capacity. Interestingly, the trigger for such phenotypically plastic cell clustering (high salinity) is frequently encountered by bacteria in diverse environments ranging from marine habitats to human skin. Therefore, such clustering is expected to have important ecological implications. We further showed that this plastic capacity to form multicellular clusters is evolvable and can be assimilated rapidly to result in bacteria that obligately grow as multicellular clusters.

Our study is unique because it demonstrates not only that phenotypic plasticity can facilitate the evolution of macroscopic multicellularity, but also that it can do so in unicellular bacteria. Specifically, although a previous study showed that the assimilation of phenotypic plasticity can lead to multicellular development in algae[11], their multicellular structures contained <200 cells and remained microscopic. Moreover, a recent important study has demonstrated the mutation-driven (i.e., not plasticity-based) evolution of macroscopic multicellularity in a eukaryote (yeast), where the largest multicellular clusters comprised ~4.5 × 10^5 cells[9]. Building on this fascinating finding, we show that phenotypic plasticity can enable unicellular *bacteria* to form macroscopic clusters comprising >10^5 CFUs under high salinity. Furthermore, we successfully assimilated this plastic phenotype to form macroscopic clusters comprising >10^4

CFUs without any environmental induction. We also note that our CFU counts within clusters are likely underestimates (see *Methods*).

Since the evolved bacteria grow obligately as macroscopic multicellular clusters even in the absence environmental induction, we conclude that they have successfully evolved the first step towards the multicellularity that requires the obligate formation of undifferentiated clusters. Importantly, such obligately multicellular growth of our evolved bacteria is distinct from the facultative formation of largely planar biofilms (with limited vertical growth) upon attachment to substrate surfaces, as shown by diverse bacterial species[58], including the polymer-degrading *Vibrio splendidus*[59]. Moreover, the obligate multicellularity of our evolved bacteria is distinct from the facultative multicellularity exhibited by *Myxobacteria* and the eukaryote *Dictyostelium discoideum*.

Our bacterial clusters grow by a combination of clonal expansion and aggregation (Supplementary movie 3), which makes their mode of multicellular growth similar to that of unicellular algae (e.g., *Chlorella vulgaris and Scenedesmus obliquus*)[60]. Interestingly, the algal clusters remain microscopic after development[60], and thus they are significantly smaller than our macroscopic bacterial clusters.

The evolution of multicellularity is considered to be one of the most frequent 'major transitions' because a large diversity of ecological conditions can make multicellularity selectively favorable[5,61]. Corroborating this notion, our results suggest that owing to phenotypically plasticity, the ability to evolve undifferentiated multicellularity should be widespread among bacteria, which comprise a rather large part of the tree of life. Crucially, plastic clustering enables bacteria to avoid waiting for the selection of specific de novo mutations that make cells stick together. Instead, environmental changes (e.g., an increase in salinity) can rapidly lead to the development of multicellular phenotypes, which could then be subjected to selection. By demonstrating this 'genes as followers' mode of evolution[33], our study also highlights the role of plasticity in a major evolutionary transition. Although most studies dealing with phenotypic plasticity tend to investigate one plastic trait[62], some studies have led to powerful insights by simultaneously investigating plasticity in multiple traits, all of which belong to the same level of biological organization[63-65]. Our study makes a significant advance in this field by investigating phenotypic plasticity and its evolution at two different levels of biological organization (*collectives* of cells (Fig. 2b) and *individual* cells (Fig. 3)). An important aspect of our experiment is that it demonstrates the simultaneous evolution of plasticity in opposite directions at different levels of organization (Supplementary Fig. S7; compare Figs. 2b and 3). Specifically, at the level of cell collectives, most of the evolved lines formed multicellular clusters under both habitual and high salinity; this phenotype was ancestrally expressed *in the presence of environmental induction* (Fig. 2b). In contrast, at the level of individual cells, the evolved lines showed non-spherical cell shapes with an average circularity of ≤ 0.667 under both low and high salinity; the ancestor expressed such a cell shape (non-spherical cells) *in the absence of environmental induction* (Fig. 3). Taken together, these observations caution against forecasting an evolutionary change in phenotypes by extrapolating from the phenotypic plasticity shown by the ancestor.

Although both spherical and rod-shaped cells can form multicellular clusters under high salinity, our selection for greater clustering under progressively reducing environmental induction ended up selecting for elongated bacterial cells (Fig. 3). Moreover, we found that all the six clones that showed highly elongated (filamentous) cells under high salinity (S2, S4, S5, R1, R2, and R3) also exhibited efficient genetic assimilation of the multicellular clustering (Fig. 2b). On the other hand, the two clones which could not assimilate multicellular clustering successfully (S3 and R4) also lacked highly elongated cells under high salinity (Fig. 3). Thus, cellular elongation under high salinity closely corresponded with the genetic assimilation of the ancestrally

plastic cell clustering. This notion aligns with two recent eukaryotic studies which argue that greater cell elongation leads to more efficient packing within clusters[9,66]. It may also explain why the assimilation of multicellularity was based on mutations predominantly in cell shape modulating peptidoglycan biosynthesis loci (Fig. 4). The highly parallel molecular evolution we observed at the level of loci points towards a putative pleiotropy between cell shape and clustering. This notion is strengthened by our observation that in clone S4, a single MraY mutation could not only assimilate the ancestrally plastic cell clustering but also give rise to highly elongated cells (Figs. 2b and 3). Moreover, despite superficially resembling *Pseudomonas fluorescens* mats formed under static conditions, the mats formed by R1, R2, R3, and R5 under optimal salinity were genotypically different: Unlike *P. fluorescens* mats that are predominantly formed by mutants overproducing cyclic-di-GMP[67], all our *E. coli* clusters were primarily caused by mutations in peptidoglycan biosynthesis genes (Figs. 2 and 4). Interestingly, in addition to an MraY mutation (linked to peptidoglycan biosynthesis), R5 also contained two mutations linked to cyclic-di-GMP expression (Supplementary Data 1). Previous experiments where *E. coli* adapted to LB under shaken conditions did not find mutations in the genes linked to cell wall assembly that mutated here (*mraY, mrdA, mrdB, mreB, murF, mppA, mrcB*); moreover, they did not report any macroscopic clustering[68,69]. Furthermore, the macroscopic clustering phenotypes found in our experiment were also not reported in an evolution experiment with *E. coli* adapting to diluted LB under resting conditions[70]. Hence, the phenotypic and genotypic changes observed in our experiment are unlikely to be primarily driven by adaptation to LB (under either resting or shaken conditions).

Do plasticity assimilating mutations get constrained by and largely occur in the metabolic pathways underlying the original induced phenotype? While comprehensively resolving this fundamental question remains challenging, we offer an important step in this direction. Specifically, we show that the biochemical basis of the assimilated phenotype (both peptide and β−1,4-glycosidic linkages) can be distinct from that of the original induced phenotype (β−1,4-glycosidic linkages but not peptide linkages; Supplementary Figs. S8–S10). Interestingly, the mutations that assimilated macroscopic multicellularity in our study occurred largely in genes linked to peptidoglycan biosynthesis (Fig. 4). Although peptidoglycan contains both β−1,4-glycosidic and peptide linkages, it resides within the periplasm (not on the cell surface) and the outer membrane precludes its hydrolysis by extracellular enzymes. Hence, the mutated genes that assimilated the ancestrally plastic multicellularity were likely unrelated to the expression of both extracellular polysaccharides with β−1,4-glycosidic linkages (found in both induced and uninduced clusters) *and* extracellular biomolecules with peptide linkages (found only in the uninduced clusters). Thus, our study suggests that plasticity assimilating mutations need not occur in genes underlying the original induced phenotype.

Apart from adding multiple key insights to the current understanding of how multicellularity evolves, our results should also act as stepping-stones for new theoretical and empirical studies in several diverse fields of inquiry (Supplementary Fig. S11). For example, why bacteria tend to form a single columnar cluster under high salinity instead of multiple globular clusters is a fascinating biophysical puzzle. Moreover, a generic tendency to form environmentally induced clusters could significantly impact the ecological interactions between multiple different bacterial species, potentially facilitating long-term co-existence by providing spatially segregated growth. Furthermore, the cells at the cluster's periphery inevitably face a different environment as compared to those at the core. Hence, an exciting new line of work would be to test if such ecological differences can drive the evolution of cellular differentiation. Finally, by demonstrating that bacteria can rapidly evolve macroscopic multicellularity, our results call for a reconsideration of why multicellular organisms are predominantly eukaryotic.

## Methods

### Bacterial strains and nutrient media

We used the following bacteria for studying the phenotypic plasticity of cell clustering: *Escherichia coli* K12 substr. MG1655 (Eco galK::cat-J23101-dTomato); *Staphylococcus aureus* JE2; *Pseudomonas aeruginosa* PAO1; *Citrobacter freundii* ATCC 8090; *Serratia marcescens* BS 303. The bacteria were cultured in liquid environments containing Luria Bertani broth (10 g/L tryptone, 5 g/L yeast extract) with 5 g/L NaCl (habitual salinity) or 60 g/L NaCl (high salinity).

### Timelapse movies

We used Canon Rebel T3i (Canon Inc. (Ōta, Tokyo, Japan)) to capture macroscopic images and then stitched them into timelapse movies using Persecond for Mac version 1.5 (Flixel Inc. (Toronto, Canada)). For all the timelapses reported in our study, we used a remote control to automatically capture an image every 4 min and published the movie files at 16 fps.

### Experimental evolution

We conducted experimental evolution with bacterial populations derived clonally from a single *E. coli* MG1655. We propagated five independent replicate populations each belonging to two distinct selection lines (S (Shaken at -180 rpm) and R (Resting)) by culturing bacteria in glass tubes containing 5 ml LB (Fig. 2a). In the beginning of the evolution experiment, the bacteria were cultured in Luria Bertani broth supplemented with 6% NaCl (w/vol). The NaCl concentration in the nutrient medium was progressively reduced over 50 days during the experiment. We periodically tested if a random subset of the evolving populations could cluster in an environment with a 1% lower salinity. If the majority of samples from this assayed subset could form macroscopic clusters, we reduced the salinity of the culture environments in our evolution experiment by 1% for all the evolving S and R populations. Thus, all the ten evolving populations (5 replicates each of S and R) experienced the same salinity on any given day. This resulted in the following [NaCl]: 6% w/vol (days 1–11), 5% w/vol (days 12–15), 4% w/vol (days 16–19); and 3% w/vol (days 20–50). We subcultured bacteria into fresh nutrient medium every 24 h using a selection protocol designed to enrich cell clustering phenotypes in the face of progressively reducing environmental induction. For each subculture, we picked a small piece of the previous day's bacterial cluster fitting within 20 μl and washed it serially in 2 ml fresh media in four distinct wells. This diluted the planktonically growing bacteria by $10^{-8}$-fold while keeping the clustered bacteria undiluted. We stored periodic cryo-stocks for all the 10 independently evolving populations. We streaked the endpoint cryo-stocks on Luria agar without any externally supplemented NaCl and isolated a colony from each population after 18 h. We used these colonies (clones S1, S2, S3, S4, S5, R1, R2, R3, R4, and R5) to conduct growth assays and genomic sequencing.

Since we always subcultured a cluster fitting within a 20 µl aliquot, the number of subcultured bacteria is unlikely to differ by more than an order of magnitude across replicates. Moreover, the expected number of subcultured bacteria was ≥$10^4$ for all of our experimental populations. Such subcultures with mean ≥$10^4$ and differences across replicates <10-fold are expected to lead to efficient and repeatable selection in asexual populations[71]. Aligning with this expectation, we also found highly parallel evolution in terms mutated loci (Fig. 4).

### Determining the relative allocations to multicellular clusters and planktonic growth

We grew clonal ancestral and evolved samples in the habitual and high salinity environments and sampled a 10 μl aliquot from the broth while deliberately avoiding macroscopic clusters. This led to the number of colony forming units (CFUs) in the planktonic phase of each culture. Next, we vortexed the culture tubes vigorously for 20 s to break the macroscopic clusters and obtain a uniform bacterial suspension in each culture tube. Immediately after vortexing, we again sampled a 10 μl aliquot from the uniform suspension to obtain the second set of CFU counts. This second set led to the total CFU counts within the culture tube. The difference between the second and the first CFU counts (i.e., the increase in CFUs observed due to the breakage of multicellular clusters by vortexing) reflected the fraction of bacteria within multicellular clusters. We conducted this assay after 24 h and 48 h under habitual and high salinity, respectively.

### Microscopy and cell shape analysis

We performed both brightfield and fluorescence microscopy with clonal ancestral and evolved samples at 100× magnification (oil immersion) using Nikon Eclipse 90i (Nikon Inc. (Amstelveen, NL)). All the samples subjected to microscopy were streaked on fresh Luria agar from their respective cryo-stocks. A single colony was then used to inoculate the liquid media in question (high versus habitual salinity) to obtain the phenotype at the level of cell collectives. 5 µl samples from fully grown liquid cultures (containing planktonic cells and/or macroscopic clusters) were spotted on a glass slide and protected with a glass coverslip, which resulted in the flattening and disintegration of the clusters. We used the Texas Red optical filter (excitation: 562/40 nm; emission: 624/40 nm) to observe cells expressing dTomato. Overlays between brightfield and fluorescent images were used to identify cell shapes and boundaries. We used the open-source software FIJI (ImageJ 1.53) for Mac to analyze cell shapes by manually tracing the cellular boundaries. We computed cellular perimeter and circularity ($= 4\pi \times \frac{area}{perimeter^2}$) using built-in functions in FIJI. The cellular perimeter was defined as the total length of the closed path encompassing the boundaries of a cell in a 2d image.

### Enzymatic assays

We used clonal ancestral and evolved samples to assay the ability of two distinct hydrolytic enzymes, cellulase (Tokyo Chemical Industries Co. Ltd., Tokyo (Japan)) and proteinase K (Thermo Scientific™), to inhibit/reduce multicellular macroscopic clustering. Specifically, we streaked the clones derived from their respective cryo-stocks on Luria agar and used a single colony to inoculate liquid cultures in six different environments: 5 ml LB with habitual or high salinity containing either cellulase (-386 U (22.7 mg)), proteinase K (-12 U (400 µg)), or no hydrolytic enzyme. We documented the phenotype at the level of cell collectives (macroscopic multicellular clusters versus turbid planktonic growth) at saturation.

### Statistics

**Cell shape plasticity.** We used two-tailed *t*-tests (unequal variance across types) to analyze the difference between cell shape parameters for a given genotype across habitual versus high salinity ($N = 40$ cells). The two cell shape parameters (perimeter and circularity) were analyzed separately.

**Viable cell allocation to clusters.** We used single simple *t*-tests to compare the evolved clones with the ancestor in terms of the fraction of viable cells found within clusters, both under high and habitual salinity ($N = 5$ distinct clones each for S (S1, S2, S3, S4, and S5) and R (R1, R2, R3, R4, and R5)). The CFU counts derived from clusters are likely underestimates because vortexing may not result in the complete breakage of multicellular clusters into single cells.

Plots: We used RStudio (version 2022.02.3 Build 492) for plotting all the quantitative data reported in our manuscript.

### Whole genome sequencing

Genomic DNA from single colonies from each population was isolated using GeneJet Genomic DNA Purification kit (Thermo Scientific™) for

whole genome sequencing on the evolved *E. coli* clones and their common ancestor. We did not conduct genomic sequencing with any other bacterial species in this study. We used a standard miniaturized protocol to prepare DNA libraries using the NEBNext Ultra II FS DNA Library Prep Kit for Illumina (New England BioLabs Inc. (Ipswich, MA, USA))[72]. The quantity of the prepared DNA libraries was validated with a Qubit© 2.0 Fluorometer (Thermo Fisher Scientific Inc. (Waltham, MA, USA)). We used the MiSeq system (Illumina Inc. San Diego, CA, USA) to perform 250-bp paired end next generation sequencing on the prepared libraries at a minimum coverage of $10\times$ (the average coverage of the detected mutations was $43.80\times$). We analyzed the sequencing output using the Geneious Prime software for Mac (v2022.0.2) and trimmed the sequencing output data using BBDuk to remove reads <20 bp or with a quality score <20. Since we conducted sequencing on clones, to avoid interpreting sequencing errors as mutations, we restricted our analysis to variants with frequencies ≥70%.

### Locating mutations on 3D protein structures

Since the crystal structure of MraY is not yet known for *E. coli*, a publicly available homology model made by Alphafold v2.0[73] was used (accession P0A6W3). We used the UCSF ChimeraX software[74] for Mac (https://www.rbvi.ucsf.edu/chimerax/) to identify and highlight the locations of the sites mutated in MraY in our experiment. The highlighted output was used to make Fig. 4b.

### Reporting summary

Further information on research design is available in the Nature Portfolio Reporting Summary linked to this article.

## Data availability

All the relevant source data are provided as a Source Data file. All relevant data on the detected mutations are provided as Supplementary Data 1. The whole genome sequences reported in this study are available from the NCBI database (accession number: PRJNA880543). Source data are provided with this paper.

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

## Acknowledgements

We thank Eric Libby, Jennifer Pentz, Anthony Sun, and Shraddha Karve for valuable discussions and constructive critiques. Y.C. was supported by a postdoctoral fellowship awarded by the Wenner-Gren Foundations (Sweden): Grants UPD2020-0113, UPD2021-0182. The funders had no role in study design, data collection and analysis, decision to publish, or preparation of the manuscript.

## Author contributions

Conceived the original idea and designed the project: Y.C. Supervised the project: P.A.L. Conducted the experiments and data analysis: Y.C. Wrote the manuscript: Y.C. and P.A.L. Acquired funding: Y.C. and P.A.L. Refined the idea and provided key critiques: S.D.

## Funding

## Competing interests

The authors declare no competing interests.
