## [Peer Review File · Nature Communications]

Bacteria evolve macroscopic multicellularity by the genetic assimilation of phenotypically plastic cell clusteringREVIEWER COMMENTS

Reviewer #1 (Remarks to the Author):

The authors report results working with an experimental bacterial evolution system in which an inducible plastic phenotypic response is canalized by genetic mutations. The canalization of phenotypic plasticity is an important area of investigation, and a tractable system for investigating it is a real advance for the field. This work is very exciting. Even so, there are some essential questions that the authors need directly address to interpret their exciting results. These questions include what constitutes canalization and how this work contributes to the multicellularity literature. To be clear, these critical questions are in support of the exciting work.

(1) The authors are motivated to conduct this study in part to distinguish mutation-led adaptation from adaptation that is the product of canalized phenotypic plasticity. The plasticity-led condition is provided by inducing an aggregating phenotype and then gradually reducing the induction factor (ie, high salinity) in a resting culture where aggregation is predicted to be adaptive. However, their experiments do not include a condition in which their population would adapt to resting culture without the induction of the plastic phenotype. The previous work that they cite about adaptation to resting cultures uses *Pseudomonas fluorescens* as the model organism, so we do not seem to have a basis for comparison in *E. coli* as to how the plasticity-led adaptation compares to mutation-led adaptation. To summarize, without a control condition in which replicate populations evolve in resting culture, it is difficult to compare and contrast the adaptations that appear in mutation-led and plasticity-led evolution. In consequence, while we see changes in phenotype as their cultures evolve, we are uncertain if the induction of the plastic phenotype accounts for any of these changes or if they would have appeared anyway in bare consequence of adapting to the resting culture. As a result, their claim that the phenotypic and genetic changes observed in these populations have distinctive features from plasticity-led evolution is uncertain.

(2) The authors claim that the persistence of the aggregating phenotype as salinity is withdrawn in the resting culture populations (R-populations) is explained because the phenotype is adaptive in resting culture. This claim is plausible in light of the extensive literature on the *P. fluorescens* "wrinkly spreader" phenotype which readily evolves in resting cultures and involves aggregations of cells. However, aggregation persists in several of the populations that evolve in shaking culture (S-populations). This undercuts the claim that the phenotype that evolves in the R-populations can be attributed to adaptation to resting conditions. The authors could demonstrate the superior adaptation of the R-populations by competing them with strains from S-populations under resting conditions. Since the conclusion that the aggregative phenotype is canalized because it is adaptive is a central claim of this study, the premise that the phenotype that evolves in the R-populations is, in fact, adaptive should be more strongly presented.

(3) Canalization of plastic phenotypes is important as a concept because of the idea that some of the variation that contributes to evolution might be supplied by plasticity in response to environmental or developmental inputs rather than by pre-existing genetic variation. However, the mechanism of canalization is conceptualized as genetic mutations that fix an adaptive phenotype previously realized only in the presence of induction cues. This requires some discussion of the relationship between the metabolic pathways that control the plastic response and the genetic pathways that fix the phenotype. For example, in *P. fluorescens* extra-cellular matrix production is controlled in the wild type by the cGMP signaling pathway which becomes constitutive in the wrinkly spreader mutant (as the authors note). In this sense then, the wrinkly spreader phenotype could be argued to result from the canalization of an inducible response in the wild type. The authors devote discussion to the *MraY* gene which is mutated in a large number of their R-clones, but there is not any evidence presented that *MraY*, or any of the genes implicated in mat formation that they also discuss, are involved in the original phenotypic plasticity involving salinity-induced aggregation. Again referring to the wrinkly spreader, the dissection of the genetic pathways involved in producing the phenotype was a large project published over several years in a series of papers, and for this reason it may be unreasonable to expect the authors to explore this subject fully before publication of this particular work. However, whether the function of phenotypic plasticity is simply to provide phenotypic variation that can be fixed by mutations to whatever

pathways, related or not, or whether the specific pathways involved in the plasticity provides the targets that lead to the canalized phenotype is an important question. Since the claim is that this experimental microbial evolution system can be used to model the transition between phenotypic plasticity and canalized phenotype, the authors should address their ability to distinguish these cases at some level.

(4) As the authors note on lines 31-32, the importance of multicellularity conceived of as a major transition in evolution is that it creates new targets for natural selection to act on. However, such new targets do not appear in the authors' system and based on other work in multicellular systems, it is not clear how new targets could emerge here. In undifferentiated cellular aggregations such as appear in this work, cells carrying new mutations must compete with cells carrying the background genotype to become established, even if the new mutation would be ultimately able to promote the reproduction of the entire aggregation. For this reason, the target of the evolutionary processes are still the individual cells, rather than the aggregation. While the authors never claim to have produced new targets, there is reason to think that simple spatial aggregation, while plausibly a necessary first step in the appearance of multicellularity, is not a particularly important step to produce higher-level organization. Growing cells on solid media so that their immobility forces them to live in closely packed colonies has not produced higher-level organization for evolution to target despite an extensive history. Moreover, the work of Hammerschmidt et al (2014) with bacterial mats that the authors cite required the researchers to perform extensive interventions during the life cycle to artificially select phenotypes in order to control conflicts in their system. Previous work by the Rainey group (Rainey & Rainey 2003) demonstrated that *P. fluorescens* mats growing at the air-liquid interface are not able to control conflict, and the mats reported in this work would seem to lack this capacity as well. For these reasons, although the authors are presenting a phenotype that involves the physical aggregation of cells, this work does not contribute to the literature on multicellularity.

To summarize, this work has great potential to explore the evolutionary canalization of phenotypic plasticity. However, whether canalization simply consists in the fixation of a phenotype that was previously inducible or whether the metabolic pathways that regulate and produce the phenotype are also involved in the evolution should be made clear. Central important claims about the canalized phenotype resulting from induction of the phenotype in the wild-type ancestor during evolution are left uncertain, as are claims that the phenotype is canalized because it is adaptive in a new environment. Additionally, some interpretation of the importance of physical aggregation on the evolutionary transition to multicellularity should be made clear.

Reviewer #2 (Remarks to the Author):

This paper shows nascent prokaryotic multicellularity evolving from the canalization of plasticity. This paper could be an important contribution to our understanding of the role of plasticity in evolution and the role of canalization in multicellularity.

Major comments:

Overall the paper is framed as 1) partly trying to understand the evolution of multicellularity in prokaryotes, and 2) partly trying to understand the role of assimilation (see below for a discussion of terms) of phenotypic plasticity in the evolution of multicellularity.

The first point is well-taken, but in the discussion, there is little comparing these results with similar experiments using eukaryotes (what did we learn from this experiment that we had not learned from eukaryotes, or from work on bacterial biofilms, myxo, or aggregates from polymer degrading bacteria?). This argument could be strengthened.

In my opinion, the second point is even more interesting: does phenotypic plasticity facilitate the evolution of innovation? Does phenotypic expression affect the tempo and mode of evolution? Thus, I would suggest changing the introduction. As it stands, each paragraph covers many topics, and the transitions are not always clear. Instead, I would talk about the importance of plasticity in novelty, the many cellular traits that have been co-opted in the evolution of multicellularity, and the importance of plasticity assimilation on the evolution of multicellular development (see for

example Aurora Nedelcu's work on co-option of plasticity on the evolution of *Volvox* differentiation).

Nevertheless, as the paper currently stands, it is hard to evaluate the importance of canalization of previously present plasticity vs. a novel canalized phenotype on the evolution of multicellularity. For example, in Figure 2 most of the S tubes (irrespective of the environmental condition) evolved multicellular phenotypes (even if the multicellular phenotypes show clear differences between low and high salinity). A detailed quantitative description of the different phenotypes (e.g., clump size distributions, and biofilm production) could provide a better framework to compare the low and high salinity treatments at the cluster level (similar to the cell level characterization). Moreover, to address the questions of canalization of plasticity and its role in adaptation, it would be important to compare the mutations of low and high salinity levels.

Overall, I think the paper could benefit from a more careful definition of canalization. Canalization (in the Waddingtonian sense) is only developmental robustness to environmental conditions (the opposite of plasticity). In this paper, canalization seems to refer to the process of increased genetic assimilation of a plastic trait (see the extensive literature on plasticity, assimilation, and evolution). I am not a purist of word use, but it would be helpful to state your definitions clearly in the introduction. This lack of clarity of terms complicates the explanations of one of the most exciting results: increased assimilation of multicellular phenotypes does not seem to be matched by assimilation of cellular plasticity. Instead, the evolution of novel cellular phenotypes follows the opposite direction of environmental plasticity. In line 237 you talk about canalization of the cellular traits in the sense of developmental or phenotypic robustness, but using the same concept downplays how different it is from the multicellular case. Even more interesting is that (as you state in line 370), unlike the ancestor, there is an association in the evolved lines between cellular elongation and canalization of the multicellular phenotype. I would love to see the quantitative relationship between cellular and multicellular phenotypes as part of the results. Finally, I got confused by the description of these results in the discussion (l 360 to 365). You state that "the ancestor expressed such cell circularity in the absence of environmental induction" but the ancestor is more circular in the saline environment.

Finally, I am concerned about the possible effects of plasticity on the effectiveness of selection due to changes in population sizes (i.e. the number of cells is likely to be variable between chunks of the clump to transfer).

Minor comments:

The second paragraph of the introduction could be streamlined. For example, you could move the sentence in line 47 after "organisms in this context" (l45). That is "...organisms in this context. For example, a recent experimental evolution experiment."

Add something to the name to distinguish high and low salinity treatments (instead of just S and R). F

L49 - Remove interestingly; it could be "despite that ..."

L69- Incomplete sentence. Separate "is the basis of..." as its own complete sentence.

L. 125 - Why is this in another sentence (why the particular grouping of bacteria?) You could merge results and refer to both figure 1 and figure s2 in the parenthesis.

L. 421 - Why are there so many nested parentheses?

L. 426 - How was the reduction rate in salinity decided?

L 237 (and elsewhere) - Maybe just say X out of Y lines and have the names in parentheses.

L. 290-292 - Again, it would be great to have a quantification of plasticity (to measure canalization).

Reviewer #3 (Remarks to the Author):

In this experimental study, Chavhan and colleagues successfully explore the role of environmental perturbation, phenotypic plasticity, and experimental evolution on the evolution of multicellular group formation in different bacterial species.

I agree with the authors that this study addresses a critical gap in the knowledge: it informs us about a particular environmental condition (i.e., high salinity) that can trigger multicellular group formation in distinct bacterial species. They then use experimental evolution to show that such a piece of information can be used to make successful predictions, which leads to the canalization of a phenotypically plastic state to genetically driven multicellularity. Furthermore, I appreciate the data presented on cell-level microscopy analysis and genome sequencing, making it a complete scientific work overall. Finally, I also agree with the authors regarding the importance of using prokaryotic species as the basis of this work.

While the study already has great scientific structure/content and is suitable for publication, I can think of a few straightforward experiments that would still improve this work.

Major suggestions:

1) In the evolved populations, cellular elongation seems to underlie multicellular cluster formation. However, the ancestral bacteria can also form macroscopic cell clusters under high salinity without needing elongated cell morphology (see spherical cells in Fig 3 top panel and compare that to the corresponding image in Fig2). This is most likely a far-fetched connection, but your result is reminiscent of Gulli et al. 2019 (<https://doi.org/10.1111/evo.13727>). The authors of that paper observed a similar, filamentous multicellular group formation in liquid culture. They discovered that cluster aggregation was driven by elevated cell death, causing the lysis of extracellular proteins, which served as molecular adhesives bringing cells together. Can similar mechanisms be at play with the ancestral bacterial cultures presented in this work? That is, high, lethal doses of salt would increase cell death and release proteins from lysed cells, resulting in the aggregation of filamentous clusters. One can quickly test this by staining ancestral populations for death cells and examining the composition of the extracellular environment in those filaments by chemical dyes. If this is incorrect, ruling out this possibility would raise interesting questions regarding the mechanisms behind group formation in the ancestor with spherical cells.

2) In the supplementary movies, the authors show growth at a macroscopic level and complement those nicely with microscopy images of single cells (Fig3). On the other hand, because cell-level images in Fig3 only show disintegrated clusters, it is unclear how those cells are organized to form macroscopic multicellular groups. For example, can the authors image two intact macroscopic clusters extracted from the S1 line with two distinct macroscopic morphologies more closely (see the two culture tubes of S1 in Fig2B)? Are there broad differences in the organization of individual cells making up the two multicellular phenotypes? Such intermediary imaging data will bridge the gap between the images of whole clusters seen from a distance (e.g., Fig1-2) and pictures of single cells as seen through microscopy images (Fig3). A couple of microscopy images showing just the surface of two unbroken multicellular clusters would be sufficient - no need to do a 3D topological examination.

3) I enjoyed reading your explanation regarding the results from genome sequencing, which makes logical assumptions about the link between peptidoglycan biosynthesis genes, cell shape, and multicellular phenotype. Nevertheless, since there are only 1-3 mutations per isolate (Table S3) and it is easy to do genetic engineering in bacteria, it makes sense to do a simple forward-genetics experiment, providing empirical evidence to connect genes to cell shape to multicellular groups directly. So why not just pick 2-3 candidate genes, or just that single mutation in clone S4, engineer them in the ancestor, and examine cell-level changes and the resulting multicellular phenotype? This forward genetics experiment should take, at most, a few weeks and provide an

irrefutable link between genes to phenotypes, at both cell- and group-level.

Minor points:

- 1) It may be interesting to briefly discuss how the nature of the difference in outer layers of gram-positive vs. gram-negative bacteria would impact the capacity to form multicellular clusters, considering that your gram-negative bacterium (*S. marcescens*) did not exhibit cluster formation in the high-salinity environment. However, if this is too speculative, please ignore this point.
- 2) Please briefly define 'cell perimeter' in the results, as it can only be intuitive to some.
- 3) Please consider moving TableS3 to the main text as part of Fig 4. That is because it summarizes genetic changes in each population better and more completely than what Fig4b tries to achieve.
- 4) In the methods, please mention that you have sequenced *E. coli*, and not any other bacterial species used in the manuscript.

RESPONSE TO REVIEWERS' COMMENTS

Response to Reviewer 1

Reviewer #1 (Remarks to the Author):

The authors report results working with an experimental bacterial evolution system in which an inducible plastic phenotypic response is canalized by genetic mutations. The canalization of phenotypic plasticity is an important area of investigation, and a tractable system for investigating it is a real advance for the field. This work is very exciting. Even so, there are some essential questions that the authors need directly address to interpret their exciting results. These questions include what constitutes canalization and how this work contributes to the multicellularity literature. To be clear, these critical questions are in support of the exciting work.

Thank you for providing constructive comments on the manuscript and sharing our excitement about its results. As advised, after conducting multiple follow-up experiments, we have made the required changes in the manuscript to address the questions raised here.

(1) The authors are motivated to conduct this study in part to distinguish mutation-led adaptation from adaptation that is the product of canalized phenotypic plasticity. The plasticity-led condition is provided by inducing an aggregating phenotype and then gradually reducing the induction factor (ie, high salinity) in a resting culture where aggregation is predicted to be adaptive. However, their experiments do not include a condition in which their population would adapt to resting culture without the induction of the plastic phenotype. The previous work that they cite about adaptation to resting cultures uses *Pseudomonas fluorescens* as the model organism, so we do not seem to have a basis for comparison in *E. coli* as to how the plasticity-led adaptation compares to mutation-led adaptation. To summarize, without a control condition in which replicate populations evolve in resting culture, it is difficult to compare and contrast the adaptations that appear in mutation-led and plasticity-led evolution. In consequence, while we see changes in phenotype as their cultures evolve, we are uncertain if the induction of the plastic phenotype accounts for any of these changes or if they would have appeared anyway in bare consequence of adapting to the resting culture. As a result, their claim that the phenotypic and genetic changes observed in these populations have distinctive features from plasticity-led evolution is uncertain.

Since we observed the genetic assimilation of the ancestrally plastic bacterial clustering after evolution under *both shaken and resting conditions*, it is unlikely that such assimilation is the mere consequence of adapting to resting culture.

We elaborate this point in more detail below and have made targeted changes in the revised manuscript to clarify this aspect; we also made changes to better describe our experimental protocol early in the revised manuscript (lines 98-100).

Lines 157-165 (revised manuscript):

"We used a single E. coli MG1655 colony to propagate two distinct experimental evolution lines (S (for Shaken) and R (for Resting)) to artificially select for increased macroscopic cell clustering in

environments with progressively reducing salinity (Fig. 2a; see Methods). Specifically, for both S and R lines, we picked a small portion of the previous day's bacterial cluster fitting within a 20 μ l aliquot and washed it four successive times in sterile 2 ml media before transferring it to a fresh culture tube. This diluted the planktonic bacteria by at least 10^8 -fold while keeping the macroscopically clustered bacteria undiluted. Thus, macroscopic clustering was favorable in both shaken and resting conditions in our artificial selection scheme."

Hence, the preferential weeding out of planktonic bacteria due to serial dilution was the principal source of selective pressure, under both shaken (for S lines) and resting (for R lines) conditions. It is possible that clustering could be adaptive also during growth, for example by increasing access to oxygen under resting conditions or reducing the effects of salt stress, but this is not required for clustering to be selected due to this strong artificial selection step.

We found that the ancestrally inducible macroscopic cell clustering was genetically assimilated under both shaken (lines S1, S2, S4, and S5) and resting (lines R1, R2, R3, and R5) conditions (Fig. 2b; also see the newly added Fig. S5 for a quantitative analogue of the phenotypes shown in Fig. 2b).

In lines 168-174 (revised manuscript), we state that...

"The rationale behind conducting experimental evolution in both resting and shaken conditions is that these two environments can select for qualitatively different clustering. This is because in resting cultures, oxygen supply depletes steeply from the air-liquid interface to tube's floor. Hence, selection for increased clustering in resting cultures is likely to enrich mutants that cluster preferentially at the air-liquid interface⁴⁸. In contrast, such oxygen availability gradients are much weaker in shaken tubes, where selection for greater clustering may not enrich interface inhabiting mutants."

In a hypothetical case, if we include additional controls without the environmental induction but with the same artificial selection scheme, there will have to be two such controls (one shaken and the other kept resting). However, our selection scheme is designed to preferentially subculture *macroscopic* clusters while avoiding turbid growth. Since there would be no *macroscopic* clusters suitable for subculturing in the absence of environmental induction, selection for multicellular phenotypes would not proceed in any of the two controls. The growth dynamics in cultures without salt would also be very different, because high salt tends to reduce the growth rate. Hence, it would be an unsuitable control for an intended single factor (+/- salt). Any differences between such controls and our treatments would thus not be interpretable in a useful way as the two controls would lead to two completely different experiments and not simply a test of plasticity-led adaptation compared to mutation-led adaptation.

In lines 473-480 (revised manuscript), we state the following:

"Previous experiments where E. coli adapted to LB under shaken conditions did not find mutations in the genes linked to cell wall

assembly that mutated here (mraY, mrdA, mrdB, mreB, murF, mppA, mrcB); moreover, they did not report any macroscopic clustering^{67,68}. Furthermore, the macroscopic clustering phenotypes found in our experiment were also not reported in an evolution experiment with E. coli adapting to diluted LB under resting conditions⁶⁹. Hence, the phenotypic and genotypic changes observed in our experiment are unlikely to be primarily driven by adaptation to LB (under either resting or shaken conditions)."

Taken together, including additional controls without environmental induction is unlikely to add to the current study (please also refer to the response to the next point).

(2) The authors claim that the persistence of the aggregating phenotype as salinity is withdrawn in the resting culture populations (R-populations) is explained because the phenotype is adaptive in resting culture. This claim is plausible in light of the extensive literature on the *P. fluorescens* "wrinkly spreader" phenotype which readily evolves in resting cultures and involves aggregations of cells. However, aggregation persists in several of the populations that evolve in shaking culture (S-populations). This undercuts the claim that the phenotype that evolves in the R-populations can be attributed to adaptation to resting conditions. The authors could demonstrate the superior adaptation of the R-populations by competing them with strains from S-populations under resting conditions. Since the conclusion that the aggregative phenotype is canalized because it is adaptive is a central claim of this study, the premise that the phenotype that evolves in the R-populations is, in fact, adaptive should be more strongly presented.

As described above, bacterial clustering is adaptive *in both shaken and resting conditions* under our artificial selection (schematically shown in Fig. 2a). This is the reason why clustering was genetically assimilated in both S and R lines (Fig. 2b). The clustered phenotype does not necessarily need to be adaptive in the absence of the artificial selection for macroscopic clustering, although it is plausible that clustering could confer a growth advantage by allowing colonization of the air-liquid interface under static conditions or protection from salt stress. Thus, that clustering phenotype is genetically assimilated because it is adaptive is not a central claim of this study. We obviously need to explain this aspect of our work better and have clarified this in the manuscript, as described in the response to the previous comment (lines 157-165; 168-174; 473-480 (revised manuscript)).

We do not expect the R lines to be superior at clustering than the S lines, both in the presence and the absence of environmental induction.

However, we did expect that the S and R lines might evolve uninduced macroscopic clustering with contrasting properties (lines 168-174 (revised manuscript)): Specifically, the R lines were expected to evolve uninduced clustering preferentially at the air-liquid interface, but the S lines were expected to show no such spatial preference in their evolved uninduced clusters.

Our results aligned with both of these expectations (Supplementary movies 5 and 6; Figs. 2b and S4). We interpret the difference as an indication that the R lines adapted to the static

conditions in addition to the artificial selection for cell clustering, but whether this is truly the case or not does not influence the main conclusions of our study.

(3) Canalization of plastic phenotypes is important as a concept because of the idea that some of the variation that contributes to evolution might be supplied by plasticity in response to environmental or developmental inputs rather than by pre-existing genetic variation. However, the mechanism of canalization is conceptualized as genetic mutations that fix an adaptive phenotype previously realized only in the presence of induction cues. This requires some discussion of the relationship between the metabolic pathways that control the plastic response and the genetic pathways that fix the phenotype. For example, in *P. fluorescens* extra-cellular matrix production is controlled in the wild type by the cGMP signaling pathway which becomes constitutive in the wrinkly spreader mutant (as the authors note). In this sense then, the wrinkly spreader phenotype could be argued to result from the canalization of an inducible response in the wild type. The authors devote discussion to the *MraY* gene which is mutated in a large number of their R-clones, but there is not any evidence presented that *MraY*, or any of the genes implicated in mat formation that they also discuss, are involved in the original phenotypic plasticity involving salinity-induced aggregation. Again referring to the wrinkly spreader, the dissection of the genetic pathways involved in producing the phenotype was a large project published over several years in a series of papers, and for this reason it may be unreasonable to expect the authors to explore this subject fully before publication of this particular work. However, whether the function of phenotypic plasticity is simply to provide phenotypic variation that can be fixed by mutations to whatever pathways, related or not, or whether the specific pathways involved in the plasticity provides the targets that lead to the canalized phenotype is an important question. Since the claim is that this experimental microbial evolution system can be used to model the transition between phenotypic plasticity and canalized phenotype, the authors should address their ability to distinguish these cases at some level.

Thank you for pointing out this important direction to improve our manuscript.

We agree that a discussion of the metabolic pathways linked to the plastic response would benefit our study. This could shed light on whether the mutations that assimilate the ancestral plasticity tend to be associated with and constrained by the pathways involved in the plasticity itself (or, alternatively, the plastic phenotype can be assimilated by mutations in unrelated genes/pathways). As the reviewer notes, fully describing the mechanistic details of how these mutations lead to macroscopic clustering will be a major undertaking beyond the scope of this article. However, through additional experiments we have been able to demonstrate that the biochemical basis of the genetically assimilated (uninduced) clustering was distinct from that of induced clustering.

Specifically, we determined whether the cell clustering observed in the ancestor and the evolved clones was reduced/inhibited by two distinct hydrolytic enzymes:

1. Cellulase
2. Proteinase K

Lines 358-364 (revised manuscript):

“In the ancestral genotype, cell clustering (which occurs only under high salinity) could be successfully reduced by cellulase but not by

proteinase K (Fig. S8; Supplementary movie 11). Although cellulase could not completely inhibit macroscopic clustering, it discernably reduced the formation of elongated filament-like clusters and increased the turbidity of the ancestral broth (Fig. S8; Supplementary movie 11). Thus, β -1,4-glycosidic linkages (but not peptide linkages) played a key role in binding the ancestral cells within clusters under high salinity.”

Lines 365-380 (revised manuscript):

“Surprisingly, in the evolved clones, we found that the bonds keeping the cells together within clusters were qualitatively different in the presence and absence of environmental induction. Under high salinity, macroscopic clustering in the evolved clones was successfully reduced by cellulase but not by proteinase K (Figs. S9 and 10; Supplementary movie 12). Cellulase prevented the formation of elongated filament-like clusters in the evolved clones under high salinity, but it could not completely inhibit macroscopic clustering (Figs. S9 and S10; Supplementary movie 12). Hence, similar to the ancestor, clustering in the evolved clones under high salinity was also mediated by β -1,4-glycosidic linkages, but not by peptide linkages.

In contrast to the above, under habitual salinity, proteinase K completely inhibited macroscopic clustering in 7/8 genotypes with genetically assimilated clustering (Figs. S9 and S10; Supplementary movie 13). Moreover, while cellulase could not completely inhibit uninduced macroscopic clustering, it resulted in markedly reduced clusters at the air-liquid interface in 7/8 genotypes with genetically assimilated macroscopic clustering (Figs. S9 and S10; Supplementary movie 13). Thus, clustering in the evolved clones under habitual salinity was mediated by both peptide and β -1,4-glycosidic linkages.”

Please refer to the Figure below for an illustration using clone S5 (refer to Figs. S9 and 10 for all the clones):

Lines 381-387 (revised manuscript):

“Taken together, the biochemical basis of the genetically assimilated (uninduced) clustering was distinct from that of induced clustering. Thus, although macroscopic clustering was genetically assimilated as a phenotype at the level of cell collectives, the underlying lower-level phenotype (i.e., the nature of bonds keeping cells together without environmental induction) was distinct from that of the original induced phenotype. This shows that the genetic assimilation of a higher-level phenotype can be brought about by the expression of contrasting lower-level phenotypes.”

In the Discussion (lines 481-495), we state the following:

“Do plasticity assimilating mutations get constrained by and largely occur in the metabolic pathways underlying the original induced phenotype? While comprehensively resolving this fundamental question remains challenging, we offer an important step in this direction. Specifically, we show that the biochemical basis of the assimilated phenotype (both peptide and β -1,4-glycosidic linkages) can be distinct from that of the original induced phenotype (β -1,4-glycosidic linkages but not peptide linkages; Figs. S8-S10). Interestingly, the mutations that assimilated macroscopic multicellularity in our study occurred largely in genes linked to peptidoglycan biosynthesis (Fig. 4). Although peptidoglycan contains both β -1,4-glycosidic and peptide linkages, it resides within the periplasm (not on the cell surface) and the outer membrane precludes its hydrolysis by extracellular enzymes. Hence, the mutated genes that assimilated the ancestrally plastic multicellularity were unrelated to the expression of both extracellular polysaccharides with β -1,4-glycosidic linkages (found

in both induced and uninduced clusters) and extracellular biomolecules with peptide linkages (found only in the uninduced clusters). Thus, our study suggests that plasticity assimilating mutations need not occur in genes underlying the original induced phenotype.”

This preliminary result can open up an exciting line of research concerning how plasticity helps in phenotypic innovation, an area that we aim to investigate further in the future but is outside the scope of the current study.

We also want to point out that mutations in *mraY* were not only found in the R clones, but also in 4/5 S clones. This suggests that it does not contribute substantially to mat formation at the air-liquid interface but is a key gene for macroscopic clustering as demonstrated by clone S4 where it is the only mutation found (lines 324-326 (revised manuscript)).

(4) As the authors note on lines 31-32, the importance of multicellularity conceived of as a major transition in evolution is that it creates new targets for natural selection to act on. However, such new targets do not appear in the authors' system and based on other work in multicellular systems, it is not clear how new targets could emerge here. In undifferentiated cellular aggregations such as appear in this work, cells carrying new mutations must compete with cells carrying the background genotype to become established, even if the new mutation would be ultimately able to promote the reproduction of the entire aggregation. For this reason, the target of the evolutionary processes are still the individual cells, rather than the aggregation. While the authors never claim to have produced new targets, there is reason to think that simple spatial aggregation, while plausibly a necessary first step in the appearance of multicellularity, is not a particularly important step to produce higher-level organization. Growing cells on solid media so that their immobility forces them to live in closely packed colonies has not produced higher-level organization for evolution to target despite an extensive history. Moreover, the work of Hammerschmidt et al (2014) with bacterial mats that the authors cite required the researchers to perform extensive interventions during the life cycle to artificially select phenotypes in order to control conflicts in their system. Previous work by the Rainey group (Rainey & Rainey 2003) demonstrated that *P. fluorescens* mats growing at the air-liquid interface are not able to control conflict, and the mats reported in this work would seem to lack this capacity as well. For these reasons, although the authors are presenting a phenotype that involves the physical aggregation of cells, this work does not contribute to the literature on multicellularity.

Thank you for pointing out the lack of clarity in terms of how our findings contribute to the multicellularity literature.

To make this point clearer, we have now added two distinct Supplementary Movie files (Supplementary movies 9 and 10) that show how our evolved clones can successfully undergo a life cycle in which a macroscopic multicellular cluster can *spontaneously* give rise to another macroscopic cluster via the growth of small propagules upon accessing fresh nutrients (please see the screenshots from Supplementary Movie 9 below):

*Our system has a key difference as compared to the wrinkly spreader mats of *Pseudomonas fluorescens*:*

Under both induced and uninduced conditions, the evolved S clones form macroscopic clusters in the bulk of the liquid, not merely interface-inhabiting mats (lines 199-204, Fig. 2b, and Supplementary movies 5 and 7 (revised manuscript)). These clusters can grow and give rise to new clusters via fragmentation, as shown above (Supplementary movies 9 and 10). We have discussed this aspect in lines 218-224 (revised manuscript).

We do not claim to have produced new targets of selection in our study. However, the ability to undergo a life cycle in which a multicellular stage can spontaneously give rise to another multicellular stage via the propagation of small (possibly unicellular) propagules opens up the possibility that undifferentiated clusters can themselves act as units of selection. At the same time, such selection at the level of clusters would operate at a different timescale as compared to selection at the level of individual cells. For example, since our evolved clones cluster together even in the absence of environmental induction, such clustering can provide a selective advantage in the presence of future protist predators.

Another such selective advantage could be realized under hazardous environmental conditions in terms of the protection afforded to cells at the core of the cluster at the expense of cells at the periphery. We agree that such a scenario could lead to a conflict between selection at level of individual cells and selection at the level of collectives. However, it can also be a platform for the future evolution of rudimentary differentiation, a possibility that we aim to test in the future. We discuss these points in lines 503-506 (revised manuscript).

Our bacterial clusters are also different from both the *P. fluorescens* wrinkly spreader system and the snowflake yeast system (Ratcliff et al. 2012, PNAS) because they show a mixture of both clonal and aggregative multicellularity (Supplementary movie 3). Our bacterial clusters will tend to avoid conflicts across different levels of organization to the extent that their multicellularity is clonal. In contrast, our system will tend to show such multilevel selective

conflicts to the extent that cheaters can emerge and exploit the aggregative nature of our system's multicellularity. However, as shown by Hammerschmidt et al. (2014), the origin of such cheaters can itself be the basis of a life cycle. Although their experiments required several artificial interventions to complete a successful life cycle, our system does not require such interventions.

Taken together, although our experiment does not claim to have produced new targets of selection, the transition from a unicellular lifestyle to undifferentiated macroscopic clustering reported has the potential to act as a stepping-stone towards new insights in the multicellularity literature. Fig. S11 (revised manuscript) presents a schematic of these new research directions that spring forth from our results.

To summarize, this work has great potential to explore the evolutionary canalization of phenotypic plasticity. However, whether canalization simply consists in the fixation of a phenotype that was previously inducible or whether the metabolic pathways that regulate and produce the phenotype are also involved in the evolution should be made clear. Central important claims about the canalized phenotype resulting from induction of the phenotype in the wild-type ancestor during evolution are left uncertain, as are claims that the phenotype is canalized because it is adaptive in a new environment. Additionally, some interpretation of the importance of physical aggregation on the evolutionary transition to multicellularity should be made clear.

Thank you for the valuable comments and the appreciation. As described above, we have addressed all of the above points in detail in our revised manuscript.

Response to Reviewer 2

Reviewer #2 (Remarks to the Author):

This paper shows nascent prokaryotic multicellularity evolving from the canalization of plasticity. This paper could be an important contribution to our understanding of the role of plasticity in evolution and the role of canalization in multicellularity.

Thank you for the constructive comments on our manuscript and sharing our excitement about its results. As advised, we have conducted multiple follow-up experiments and made the required changes in the manuscript to address the questions raised here.

Major comments:

Overall the paper is framed as 1) partly trying to understand the evolution of multicellularity in prokaryotes, and 2) partly trying to understand the role of assimilation (see below for a discussion of terms) of phenotypic plasticity in the evolution of multicellularity.

The first point is well-taken, but in the discussion, there is little comparing these results with similar experiments using eukaryotes (what did we learn from this experiment that we had not learned from eukaryotes, or from work on bacterial biofilms, myxo, or aggregates from polymer degrading bacteria?). This argument could be strengthened.

As advised, we have made changes to the Discussion to address this concern:

Lines 412-425 (revised manuscript):

*“Since the evolved bacteria grow obligately as macroscopic multicellular clusters even in the absence environmental induction, we conclude that they have successfully evolved the first step towards the multicellularity that requires the obligate formation of undifferentiated clusters. Importantly, such obligately multicellular growth of our evolved bacteria is distinct from the facultative formation of largely planar biofilms (with limited vertical growth) upon attachment to substrate surfaces, as shown by diverse bacterial species⁵⁷, including the polymer-degrading *Vibrio splendidus*⁵⁸. Moreover, the obligate multicellularity of our evolved bacteria is distinct from the facultative multicellularity exhibited by *Myxobacteria* and the eukaryote *Dictyostelium discoideum*.*

*Our bacterial clusters grow by a combination of clonal expansion and aggregation (Supplementary movie 3), which makes their mode of multicellular growth similar to that of unicellular algae (e.g., *Chlorella vulgaris* and *Scenedesmus obliquus*)⁵⁹. Interestingly, the algal clusters remain microscopic after development⁵⁹, and thus they are significantly smaller than our macroscopic bacterial clusters.”*

This adds to the previous discussion about the uniqueness of our results (lines 400-410 (revised manuscript)):

“Our study is unique because it demonstrates not only that phenotypic plasticity can facilitate the evolution of macroscopic multicellularity, but also that it can do so in unicellular bacteria. Specifically, although a previous study showed that the assimilation of phenotypic plasticity can lead to multicellular development in algae¹¹, their multicellular structures contained < 200 cells and remained microscopic. Moreover, a recent important study has demonstrated the mutation-driven (i.e., not plasticity-based) evolution of macroscopic multicellularity in a eukaryote (yeast), where the largest multicellular clusters comprised ~ 4.5 × 10⁵ cells⁹. Building on this fascinating finding, we show that phenotypic plasticity can enable unicellular bacteria to form macroscopic clusters comprising > 10⁵ CFUs under high salinity. Furthermore, we successfully assimilated this plastic phenotype to form macroscopic clusters comprising > 10⁴ CFUs without any environmental induction.”

In my opinion, the second point is even more interesting: does phenotypic plasticity facilitate the evolution of innovation? Does phenotypic expression affect the tempo and mode of evolution? Thus, I would suggest changing the introduction. As it stands, each paragraph covers many topics, and the transitions are not always clear. Instead, I would talk about the importance of plasticity in novelty, the many cellular traits that have been co-opted in the evolution of multicellularity, and the importance of plasticity assimilation on the evolution of multicellular development (see for example Aurora Nedelcu's work on co-option of plasticity on the evolution of *Volvox* differentiation).

As advised, we have made multiple changes in the Introduction to address the above issues. Specifically, we have...

1. Improved the transitions between paragraphs.
2. Added a dedicated description of the importance of plasticity in generating evolutionary novelty (lines 66-81 (revised manuscript)):

“Phenotypic plasticity, which enables a given genotype to express different phenotypes in different environments²⁶⁻²⁸, can be an important source of evolutionary novelty throughout the tree of life²⁹⁻³¹. This is because plasticity can facilitate biological innovation by allowing genes to be ‘followers’ in the evolution of new phenotypes^{32,33}. Specifically, plastic phenotypes can be genetically assimilated when selection enriches mutations that make their expression constitutively expressed, even in the absence of environmental induction³⁴⁻³⁶. Thus, phenotypic plasticity has the potential to accelerate evolutionary innovation by bypassing the wait for mutations required for new beneficial phenotypes. Phenotypic plasticity can also play an important

role in the evolution of multicellularity. For example, environmentally induced stress responses have been co-opted to evolve the germline-soma differentiation in the multicellular alga Volvox carteri^{37,38}. Similarly, the environmentally induced cAMP-based stress response has been co-opted for multicellular development in the social amoeba Dictyostelium discoideum³⁹. While the co-option of unicellular ancestral traits/pathways has been an important theme in the evolution of multicellularity⁴⁰⁻⁴², many such unicellular traits were phenotypically plastic, and their expression has evolved from an environmentally induced (temporal) to developmental (spatial) context³⁸.”

3. Added a sentence highlighting how plasticity has the potential to accelerate evolution (lines 72-73 (revised manuscript)).
4. Added a description of the importance of plasticity co-option for the evolution of multicellularity (lines 74-81 (revised manuscript, pasted above)).

Nevertheless, as the paper currently stands, it is hard to evaluate the importance of canalization of previously present plasticity vs. a novel canalized phenotype on the evolution of multicellularity. For example, in Figure 2 most of the S tubes (irrespective of the environmental condition) evolved multicellular phenotypes (even if the multicellular phenotypes show clear differences between low and high salinity). A detailed quantitative description of the different phenotypes (e.g., clump size distributions, and biofilm production) could provide a better framework to compare the low and high salinity treatments at the cluster level (similar to the cell level characterization).

We have now conducted two distinct experiments to address the concerns raised here:

A. We assayed the ability of two distinct hydrolytic enzymes (cellulase and proteinase K) to inhibit cell clustering under both habitual and high salinity. The detailed results are presented in Figs. S8, S9, and S10 along with Supplementary movies 11, 12, and 13. We established the following (please also see the response to Point 3 raised by Reviewer 1):

1. The ancestral cells are held together in clusters (induced under high salinity) by biomolecules containing β -1,4-glycosidic linkages but not peptide linkages.
2. The ancestral and evolved clustering phenotypes use different clustering mechanisms.
3. Under high salinity, cellulase can inhibit the formation of elongated clusters in most evolved clones while failing to completely inhibit clustering.
4. Under habitual salinity, proteinase K completely inhibited clustering in 7/10 evolved clones. In these clones, cellulase tends to reduce clustering but does not inhibit it completely.

We showed that the biochemical basis of the assimilated phenotype (both peptide and β -1,4-glycosidic linkages) was distinct from that of the original induced phenotype (β -1,4-glycosidic linkages but not peptide linkages; Figs. S8-S10 (revised manuscript)).

Lines 381-387 (revised manuscript):

“Taken together, the biochemical basis of the genetically assimilated (uninduced) clustering was distinct from that of induced clustering. Thus, although macroscopic clustering was genetically assimilated as a phenotype at the level of cell collectives, the underlying lower-level phenotype (i.e., the nature of bonds keeping cells together without environmental induction) was distinct from that of the original induced phenotype. This shows that the genetic assimilation of a higher-level phenotype can be brought about by the expression of contrasting lower-level phenotypes.”

B. For each evolved clone and the common ancestor, we conducted a detailed quantitative determination of the number of bacteria present in the turbid broth versus the number of bacteria present in multicellular clusters, under both high and habitual salinity.

Lines 556-566 (revised manuscript):

“We grew clonal ancestral and evolved samples in the habitual and high salinity environments and sampled a 10 μ l aliquot from the broth while deliberately avoiding macroscopic clusters. This led to the number of colony forming units (CFUs) in the planktonic phase of each culture. Next, we vortexed the culture tubes vigorously for 20 seconds to break the macroscopic clusters and obtain a uniform bacterial suspension in each culture tube. Immediately after vortexing, we again sampled a 10 μ l aliquot from the uniform suspension to obtain the second set of CFU counts. This second set led to the total CFU counts within the culture tube. The difference between the second and the first CFU counts (i.e., the increase in CFUs observed due to the breakage of multicellular clusters by vortexing) reflected the fraction of bacteria within multicellular clusters. We conducted this assay after 24 h and 48 h under habitual and high salinity, respectively.”

The results of this analysis are presented in Fig. S5 (revised manuscript):

This analysis revealed multiple new insights:

1. The ancestral reaction norm of bacterial allocation to clusters got genetically assimilated in the evolved lines towards the induced phenotypic level (Fig. S7b (revised manuscript)).
2. The lines S3 and R4, which contained peptidoglycan mutations and also displayed cell shape elongation relative to the ancestor under high salinity, did not show any macroscopic clumps. This discrepancy was resolved by Fig. S5 (revised manuscript), which showed that S3 and R4 indeed formed clusters, but the clusters were microscopic and hence not detectable in the images of culture tubes.

As demonstrated in Supplementary movies 5, 6, 7, and 8, the macroscopic clusters develop (and, in many cases, sink) in such a highly dynamic manner that it prevents a time-stamped analysis of cluster shape/size distributions using image analysis. Instead, we have presented an analysis of the relative distribution of viable cells in multicellular clusters versus the turbid broth. We have also done quantitative analyses to compare phenotypes at the level of cell collectives to phenotypes at the level of individual cells (Fig. S7); this has been described in detail in the response to a later comment.

Moreover, to address the questions of canalization of plasticity and its role in adaptation, it would be important to compare the mutations of low and high salinity levels.

Our experiment did not have lines evolving under high versus low salinity. Instead, both S (Shaken) and R (Resting) lines experienced an identical and progressive decrease in their environmental salinity (lines 531-537 (revised manuscript)). At the end of the experiment, we assayed both S and R lines in the habitual and high salinity environments.

Hence, there were no mutations specific to low versus high salinity levels.

Overall, I think the paper could benefit from a more careful definition of canalization. Canalization (in the Waddingtonian sense) is only developmental robustness to environmental conditions (the opposite of plasticity). In this paper, canalization seems to refer to the process of increased genetic assimilation of a plastic trait (see the extensive literature on plasticity, assimilation, and evolution). I am not a purist of word use, but it would be helpful to state your definitions clearly in the introduction. This lack of clarity of terms complicates the explanations of one of the most exciting results: increased assimilation of multicellular phenotypes does not seem to be matched by assimilation of cellular plasticity. Instead, the evolution of novel cellular phenotypes follows the opposite direction of environmental plasticity.

We agree. To avoid this confusion, we have opted to only use genetic assimilation (and not canalization) throughout the manuscript. As advised, we have explicitly defined genetic assimilation in the Introduction:

- Lines 69-71 (revised manuscript): “[Specifically,] plastic phenotypes can be genetically assimilated when selection enriches mutations that

make their expression constitutive, even in the absence of environmental induction^{34–36}.”

This has also resulted in a change in the title:

New title: *Bacteria evolve macroscopic multicellularity by the genetic assimilation of phenotypically plastic cell clustering*

As advised, we have described in detail the exciting result that phenotypic plasticity evolved in opposite directions at the level of cell collectives (clusters) and individual cells (lines 274-284 (revised manuscript)):

“We further found that plastic phenotypes evolved in opposite directions at the level of cell collectives and individual cells (Fig. S7). Specifically, at the level of individual cells, the evolved cell shape under both habitual and high salinity for both S and R matched the uninduced ancestral cell shape (i.e., the evolved cells were non-spherical; Fig. S7). In contrast, at the level of cell collectives, both S and R exhibited cell clustering under both habitual and high salinity (Figs. 2 and S5). Moreover, in both S and R, the fraction of viable cells found within multicellular clusters was relatively closer to the induced ancestral value (Fig. S7). Thus, at the level of cell collectives, the evolved (and genetically assimilated) phenotype matched the induced ancestral phenotype (multicellular clustering; Fig. S7). Taken together, we showed that plastic phenotypes can evolve in opposite directions at different levels of biological organization.”

After conducting further quantitative analyses at both levels of organization, we have added a Supplementary Figure (Fig. S7) to the revised manuscript to explicitly demonstrate how plasticity evolved in opposite directions at the level of cell collectives versus the level of individual cells:

In line 237 you talk about canalization of the cellular traits in the sense of developmental or phenotypic robustness, but using the same concept downplays how different it is from the multicellular case.

As advised, in order to prevent the downplaying of the contrasts between plasticity evolution at the level of cell collectives and individual cells, we have removed the word “canalize” in the context of lack of plasticity in cell shape from line 237 of the old manuscript (lines 261-262 in the revised manuscript). Instead, we have stated the following:

“We also found that 5/10 lines (S1, S3, R1, R4, and R5) exhibited similar cell circularity across habitual and high salinities (Fig. S6; Table S2).”

Even more interesting is that (as you state in line 370), unlike the ancestor, there is an association in the evolved lines between cellular elongation and canalization of the multicellular phenotype. I would love to see the quantitative relationship between cellular and multicellular phenotypes as part of the results.

We agree that highlighting the association between the genetic assimilation of macroscopic clustering and cell shape evolution would benefit our study. However, as described in the response to a previous comment, the dynamic development of multicellular clusters prevents a time-stamped quantitative analysis of cluster shape/size distributions using image analysis.

Moreover, as shown in Fig. 3, although the evolved clones show elongated cells under *high* salinity, they do not tend to exhibit such cell elongation under *habitual* salinity. Since the genetic assimilation of clustering was established by assaying multicellular phenotypes without induction (*i.e.*, under habitual salinity), an association between cell length and the genetic assimilation of clustering should

ideally involve the reaction norm of cell perimeter, not just its value under high (or habitual) salinity. Incorporating this notion, we have added Fig. S7 to the revised manuscript to show that the ancestrally plastic cell clustering and cell shape evolved in opposite directions.

Furthermore, in lines 248-255 (revised manuscript), we state the following:

“We found a clear correspondence between reversal of the cell perimeter plasticity and successful genetic assimilation of cellular clustering: The eight lines that showed a reversal in the ancestral cell perimeter plasticity were also the ones that successfully genetically assimilated the cellular clustering during experimental evolution (compare Figs. 2 and 3). On the other hand, the remaining two lines (S3 and R4) which showed no significant difference in cell perimeters under habitual versus high salinity were also the ones that failed to successfully assimilate macroscopic clustering (compare Figs. 2 and 3).”

In lines 453-459 (revised manuscript), we note the following:

“Moreover, we found that all the six clones that showed highly elongated (filamentous) cells under high salinity (S2, S4, S5, R1, R2, and R3) also exhibited efficient genetic assimilation of the multicellular clustering (Fig. 2b). On the other hand, the two clones which could not assimilate multicellular clustering successfully (S3 and R4) also lacked highly elongated cells under high salinity (Fig. 3). Thus, cellular elongation under high salinity closely corresponded with the genetic assimilation of the ancestrally plastic cell clustering.”

Taken together, the above descriptions should better highlight how the genetic assimilation of clustering was related to the evolution of cell shapes.

Finally, I got confused by the description of these results in the discussion (l 360 to 365). You state that "the ancestor expressed such cell circularity in the absence of environmental induction" but the ancestor is more circular in the saline environment.

We have edited this section and added clarifications to avoid confusions:

Lines 443-448 (revised manuscript):

“Specifically, at the level of cell collectives, most of the evolved lines formed multicellular clusters under both habitual and high salinity; this phenotype was ancestrally expressed in the presence of environmental induction (Fig. 2b). In contrast, at the level of individual cells, the evolved lines showed non-spherical cell shapes with an average circularity of ≤ 0.667 under both low and high

salinity; the ancestor expressed such a cell shape (non-spherical cells) in the absence of environmental induction (Fig. 3)."

Finally, I am concerned about the possible effects of plasticity on the effectiveness of selection due to changes in population sizes (i.e. the number of cells is likely to be variable between chunks of the clump to transfer).

We agree the number of bacteria within period bottlenecks is expected to vary, both among the replicate populations and also for each replicate population across time points. However, such variation is unlikely to lead to differences in the effectiveness of selection across the replicate populations for the following reasons (stated in lines 548-554 (revised manuscript)):

"Since we always subcultured a cluster fitting within a 20 μ l aliquot, the number of subcultured bacteria is unlikely to differ by more than an order of magnitude across replicates. Moreover, the expected number of subcultured bacteria was $\geq 10^4$ for all of our experimental populations. Such subcultures with mean $\geq 10^4$ and differences across replicates < 10 -fold are expected to lead to efficient and repeatable selection in asexual populations⁷⁰. Aligning with this expectation, we also found highly parallel evolution in terms mutated loci (Fig. 4)."

Minor comments:

The second paragraph of the introduction could be streamlined. For example, you could move the sentence in line 47 after "organisms in this context" (l45). That is "...organisms in this context. For example, a recent experimental evolution experiment."

As advised, we have streamlined this paragraph (lines 45-50 (revised manuscript)).

Add something to the name to distinguish high and low salinity treatments (instead of just S and R). F

As stated in lines 157-160 of the revised manuscript (lines 149-150 of the original manuscript), S and R refer to Shaken and Resting lines, respectively.

Our experiment did not have lines evolving under high versus low salinity. Instead, both S and R lines experienced an identical and progressive decrease in their environmental salinity. At the end of the experiment, we assayed both S and R lines in the habitual and high salinity environments.

L49 - Remove interestingly; it could be "despite that ..."

Done.

L69- Incomplete sentence. Separate "is the basis of..." as its own complete sentence.

We have fixed this sentence.

L. 125 - Why is this in another sentence (why the particular grouping of bacteria?) You could merge results and refer to both figure 1 and figure s2 in the parenthesis.

We have decided not to merge these results because of the following reason: there is a direct link between Fig. 1 and Supplementary movies 1 – 4. Hence, they need to be mentioned together. Mentioning Fig. S2 in the same sentence can lead to confusion as there are no movies linked to Fig. S2.

L. 421 - Why are there so many nested parentheses?

We have fixed this issue.

L. 426 - How was the reduction rate in salinity decided?

The reduction rate in salinity was decided on the go during the evolution experiment. We have described this in the revised manuscript in lines 531-534:

“We periodically tested if a random subset of the evolving populations could cluster in an environment with a 1% lower salinity. If the majority of samples within this assayed subset could form macroscopic clusters, we reduced the salinity of the culture environments in our evolution experiment by 1% for all the S and R populations.”

L 237 (and elsewhere) - Maybe just say X out of Y lines and have the names in parentheses.

We have made this change.

L. 290-292 - Again, it would be great to have a quantification of plasticity (to measure canalization).

As advised, we quantified the relative fraction of viable cells found within clusters to quantify plasticity and its assimilation (Figs. S5 and S7 (revised manuscript)).

Reviewer #3 (Remarks to the Author):

In this experimental study, Chavhan and colleagues successfully explore the role of environmental perturbation, phenotypic plasticity, and experimental evolution on the evolution of multicellular group formation in different bacterial species.

I agree with the authors that this study addresses a critical gap in the knowledge: it informs us about a particular environmental condition (i.e., high salinity) that can trigger multicellular group formation in distinct bacterial species. They then use experimental evolution to show that such a piece of information can be used to make successful predictions, which leads to the canalization of a phenotypically plastic state to genetically driven multicellularity. Furthermore, I appreciate the data presented on cell-level microscopy analysis and genome sequencing, making it a complete scientific work overall. Finally, I also agree with the authors regarding the importance of using prokaryotic species as the basis of this work.

While the study already has great scientific structure/content and is suitable for publication, I can think of a few straightforward experiments that would still improve this work.

Thank you for the valuable comments and for appreciating our study. As outlined below (and in the response to the other two reviewers), we have conducted multiple experiments and have made several changes in the text to improve our manuscript.

Major suggestions:

1) In the evolved populations, cellular elongation seems to underlie multicellular cluster formation. However, the ancestral bacteria can also form macroscopic cell clusters under high salinity without needing elongated cell morphology (see spherical cells in Fig 3 top panel and compare that to the corresponding image in Fig2). This is most likely a far-fetched connection, but your result is reminiscent of Gulli et al. 2019 (<https://doi.org/10.1111/evo.13727>). The authors of that paper observed a similar, filamentous multicellular group formation in liquid culture. They discovered that cluster aggregation was driven by elevated cell death, causing the lysis of extracellular proteins, which served as molecular adhesives bringing cells together. Can similar mechanisms be at play with the ancestral bacterial cultures presented in this work? That is, high, lethal doses of salt would increase cell death and release proteins from lysed cells, resulting in the aggregation of filamentous clusters. One can quickly test this by staining ancestral populations for death cells and examining the composition of the extracellular environment in those filaments by chemical dyes. If this is incorrect, ruling out this possibility would raise interesting questions regarding the mechanisms behind group formation in the ancestor with spherical cells.

As advised, we used hydrolytic enzymes to determine the biochemical basis of phenotypically plastic clustering and its evolution.

Specifically, we tested if the ancestral bacteria and the evolved clones could develop clusters in the presence of two distinct hydrolytic enzymes:

1. Cellulase
2. Proteinase K

Lines 358-364 (revised manuscript):

“In the ancestral genotype, cell clustering (which occurs only under high salinity) could be successfully reduced by cellulase but not by proteinase K (Fig. S8; Supplementary movie 11). Although cellulase could not completely inhibit macroscopic clustering, it discernably reduced the formation of elongated filament-like clusters and increased the turbidity of the ancestral broth (Fig. S8; Supplementary movie 11). Thus, β -1,4-glycosidic linkages (but not peptide linkages) played a key role in binding the ancestral cells within clusters under high salinity.”

Lines 365-380 (revised manuscript):

“Surprisingly, in the evolved clones, we found that the bonds keeping the cells together within clusters were qualitatively different in the presence and absence of environmental induction. Under high salinity, macroscopic clustering in the evolved clones was successfully reduced by cellulase but not by proteinase K (Figs. S9 and 10; Supplementary movie 12). Cellulase prevented the formation of elongated filament-like clusters in the evolved clones under high salinity, but it could not completely inhibit macroscopic clustering (Figs. S9 and S10; Supplementary movie 12). Hence, similar to the ancestor, clustering in the evolved clones under high salinity was also mediated by β -1,4-glycosidic linkages, but not by peptide linkages.

In contrast to the above, under habitual salinity, proteinase K completely inhibited macroscopic clustering in 7/8 genotypes with genetically assimilated clustering (Figs. S9 and S10; Supplementary movie 13). Moreover, while cellulase could not completely inhibit uninduced macroscopic clustering, it resulted in markedly reduced clusters at the air-liquid interface in 7/8 genotypes with genetically assimilated macroscopic clustering (Figs. S9 and S10; Supplementary movie 13). Thus, clustering in the evolved

clones under habitual salinity was mediated by both peptide and β -1,4-glycosidic linkages.”

Please refer to the Figure below for an illustration using clone S5 (refer to Figs. S9 and 10 for all the clones):

Lines 381-387 (revised manuscript):

“Taken together, the biochemical basis of the genetically assimilated (uninduced) clustering was distinct from that of induced clustering. Thus, although macroscopic clustering was genetically assimilated as a phenotype at the level of cell collectives, the underlying lower-level phenotype (i.e., the nature of bonds keeping cells together without environmental induction) was distinct from that of the original induced phenotype. This shows that the genetic assimilation of a higher-level phenotype can be brought about by the expression of contrasting lower-level phenotypes.”

In the Discussion (lines 481-495), we state the following:

“Do plasticity assimilating mutations get constrained by and largely occur in the metabolic pathways underlying the original induced phenotype? While comprehensively resolving this fundamental question remains challenging, we offer an important step in this direction. Specifically, we show that the biochemical basis of the assimilated phenotype (both peptide and β -1,4-glycosidic linkages) can be distinct from that of the original induced phenotype (β -1,4-glycosidic linkages but not peptide linkages; Figs. S8-S10). Interestingly, the mutations that assimilated macroscopic multicellularity in our study occurred largely in genes linked to peptidoglycan biosynthesis (Fig. 4). Although peptidoglycan contains both β -1,4-glycosidic and peptide linkages, it resides within the periplasm (not on the cell surface) and the outer membrane precludes its hydrolysis by extracellular enzymes.

Hence, the mutated genes that assimilated the ancestrally plastic multicellularity were unrelated to the expression of both extracellular polysaccharides with β -1,4-glycosidic linkages (found in both induced and uninduced clusters) and extracellular biomolecules with peptide linkages (found only in the uninduced clusters). Thus, our study suggests that plasticity assimilating mutations need not occur in genes underlying the original induced phenotype.”

Having performed the above assays, we opted against determining if dead cells acted as sources of extracellular glues. This is because even if dead cells are detected in microscopy images, given the small size of a single cell as compared to the size of the macroscopic cluster, it is difficult to demonstrate that they act as sources of the extracellular glue.

2) In the supplementary movies, the authors show growth at a macroscopic level and complement those nicely with microscopy images of single cells (Fig3). On the other hand, because cell-level images in Fig3 only show disintegrated clusters, it is unclear how those cells are organized to form macroscopic multicellular groups. For example, can the authors image two intact macroscopic clusters extracted from the S1 line with two distinct macroscopic morphologies more closely (see the two culture tubes of S1 in Fig2B)? Are there broad differences in the organization of individual cells making up the two multicellular phenotypes? Such intermediary imaging data will bridge the gap between the images of whole clusters seen from a distance (e.g., Fig1-2) and pictures of single cells as seen through microscopy images (Fig3). A couple of microscopy images showing just the surface of two unbroken multicellular clusters would be sufficient - no need to do a 3D topological examination.

Thank you for this valuable suggestion. While it would be a very exciting extension of our work, we are unable to do it for the current manuscript for two specific reasons:

1. Capturing a focused image of the macroscopic multicellular clusters poses major empirical difficulties because the clusters lose their shape when the surrounding liquid is removed.
2. It is difficult to achieve focus under a microscope when the clusters are kept within their surrounding liquids (the clusters are too large, and the constituent bacterial cells are too small). This applies to both the large- and intermediate-scale magnification needed to visualize individual bacterial cells.

3) I enjoyed reading your explanation regarding the results from genome sequencing, which makes logical assumptions about the link between peptidoglycan biosynthesis genes, cell shape, and multicellular phenotype. Nevertheless, since there are only 1-3 mutations per isolate (Table S3) and it is easy to do genetic engineering in bacteria, it makes sense to do a simple forward-genetics experiment, providing empirical evidence to connect genes to cell shape to multicellular groups directly. So why not just pick 2-3 candidate genes, or just that single mutation in clone S4, engineer them in the ancestor, and examine cell-level changes and the resulting multicellular phenotype? This forward genetics experiment should take, at most, a few weeks and provide an irrefutable link between genes to phenotypes, at both cell- and group-level.

Thank you for this valuable suggestion. Since one of the evolved clones (S4) was only one *mraY* mutation away from the common ancestor, it is very likely that *mraY* played an

important the role in genetically assimilating the plastic multicellularity in our evolution experiment.

We have already undertaken a reconstruction project to better understand both pleiotropy and epistasis with respect to the mutated loci that we observed in our current experiment. However, we plan to conduct this as a detailed and dedicated study that will hopefully reveal many more insights in the future.

Minor points:

1) It may be interesting to briefly discuss how the nature of the difference in outer layers of gram-positive vs. gram-negative bacteria would impact the capacity to form multicellular clusters, considering that your gram-negative bacterium (*S. marcescens*) did not exhibit cluster formation in the high-salinity environment. However, if this is too speculative, please ignore this point.

Since we saw multicellular clusters in both Gram-negative (*Escherichia coli*, *Pseudomonas aeruginosa*, and *Citrobacter freundii*) and Gram-positive bacteria (*Staphylococcus aureus*), we concluded that both kinds of bacteria can make clusters despite the qualitative differences in their envelopes. Hence, we do not conclude that the inability of *S. marcescens* to form multicellular clusters is due to the presence of a Gram-negative cell envelope.

2) Please briefly define 'cell perimeter' in the results, as it can only be intuitive to some.

As advised, in lines 580-582, we have added the following:

“The cellular perimeter was defined as the total length of the closed path encompassing the boundaries of a cell in a 2d image.”

3) Please consider moving TableS3 to the main text as part of Fig 4. That is because it summarizes genetic changes in each population better and more completely than what Fig4b tries to achieve.

The large size of Table S3 makes it difficult to fit within the main text on a single printed page as a part of Fig. 4. Moreover, Fig. 4b also reveals important spatial information about the mutated locations on the MraY protein. Specifically, all the MraY mutations were located on the periplasmic side of transmembrane protein, away from its cytoplasmic catalytic site. This periplasmic region of MraY remains relatively understudied. Since such information is not revealed by Table S3, we would like to retain Fig. 4b in its current location within the main text while keeping Table S3 in the Supplementary Information.

4) In the methods, please mention that you have sequenced *E. coli*, and not any other bacterial species used in the manuscript.

As advised, we have made this change in lines 605-608:

“Genomic DNA from single colonies from each population was isolated using GeneJet Genomic DNA Purification kit (Thermo Scientific™) for

whole genome sequencing on the evolved E. coli clones and their common ancestor. We did not conduct genomic sequencing with any other bacterial species in this study.”

Additional changes (not mentioned in the reviewer comments)

1. We have replaced Figs. S4 and S7 of the old manuscript with a more comprehensive analysis presented in Fig. S5 of the revised manuscript.
2. To remove ambiguities, we have replaced Supplementary movie 9 of the old manuscript with two distinct Supplementary movies (9 and 10) in the revised manuscript.

REVIEWERS' COMMENTS

Reviewer #2 (Remarks to the Author):

Dear authors, first of all, I would like to apologize for taking an unreasonable amount of time to revise your manuscript a second time. I can see that you have put in a lot of work and made the paper much better. I really liked the clearer language and new experiments. Overall, I think this is an important contribution to the evolution of multicellularity literature, demonstrating the importance of previous plasticity. I think the paper is ready for publication.

Reviewer #3 (Remarks to the Author):

The authors have addressed most of my comments, and I am pleased that they intend to conduct a comprehensive study to investigate the genetic basis of filamentous group formation. I am happy with the publication of this study.

Reviewer #4 (Remarks to the Author):

The authors have submitted an improved manuscript and I remain excited about the high value of a tractable experimental system that can be used to investigate the role of phenotypic plasticity in evolutionary changes. In what follows, I will revisit the four comments I made on the author's first submission and evaluate their responses to my concerns.

(1) In my review of the original manuscript, I commented that the authors had not proven that phenotypic plasticity played a role in the physical aggregation that persisted as they withdrew salinity in the resting culture because the physical aggregations might have evolved in the absence of salinity-mediated induction. My understanding was that the authors were trying to demonstrate a process where phenotypic plasticity (ie, the clustering phenotype induced by high salinity) was scaffolding adaptation to inhabiting the air-liquid interface of the resting cultures. Based on that understanding, I interpreted the shaking cultures as a control designed to show the differences in evolution when the culture conditions did not provide a benefit (ie, increased availability of oxygen) at the air-liquid interface. I thank the authors for clarifying the process and intentions of their study.

The procedure is designed to exclude planktonically growing cells and allow only the transfer of physically aggregated biomass. The reason that the aggregated phenotype (inducible in the ancestor) is adaptive to the selected conditions is because the phenotypes most resistant to disaggregation under their washing procedure will be most likely to transfer to a new culture as salinity is withdrawn. The phenotype is adaptive under this transfer procedure in both the resting and shaken condition. Thus, we see that the genetic assimilation of the inducible phenotype proceeds under two different environments when the selective conditions are applied. The revised manuscript clarifies the purpose of their paper and their response explains why my original comment does not address their purpose.

However, these clarifications raise concerns about the validity of their procedure as it relates to the goals of their study. Rather than using phenotypic plasticity to provide variation to promote adaptation to conditions that favor multicellularity, the authors might be better characterized as demonstrating a technique for isolating mutations that make a formerly inducible phenotype into a constituent phenotype in the absence of induction. The authors characterize this process as "artificial selection." The question of whether the selective conditions imposed on a population in a laboratory experiment could be characterized as "natural" is somewhat philosophical, but for an experimental study to inform understanding of natural processes would depend on how well the laboratory selection process models natural selection outside the laboratory. The authors have not supported how a process that requires the judgement and conditional manipulation of the system by the experimenter (ie, identifying and isolating a portion of aggregated biomass to wash and

transfer) is a valid model for the selective conditions that would have led to multicellularity in nature.

While this concern does not diminish the potential of this system to investigate the role of phenotypic plasticity during evolutionary change, it reduces the applicability of these particular experiments to understanding natural processes where there is not an intelligence to guide the trajectory of evolution. There are established conditions demonstrated to promote physical aggregation that model natural selection more closely, including increased oxygen availability at the air-liquid interface (Rainey and Traviarno 1998 and subsequent work), using settling to select on particle size (Ratcliff et al 2012 and subsequent work), and predation (Boraas et al 1999 and subsequent work), and so demonstrating that variation introduced by phenotypic plasticity can contribute to the evolution of physical aggregation under any of these conditions would clearly represent an advance in the field. Introducing an unrelated artificial selection process is more ambiguous as to whether it is providing new insights into this evolutionary transition.

(2) The authors are correct that my concerns about the differences between the aggregations present in the resting and shaken cultures do not apply to their study once the procedure and purpose is properly clarified. The changes I suggested in my first review are not needed and I appreciate their discussion of the differences between the observed outcomes in the resting and shaken cultures. These differences do make it plausible that the plastic aggregation might be acting as a scaffold for adaptation to the air-liquid interface in the resting cultures.

(3) In my review of the authors' first submission, I requested a discussion of how the pathways mediating a plastic response would function in evolutionary change. Specifically, I asked for a discussion on whether the plasticity was only providing variation that could be stabilized by any means or whether the pathways that mediated the plasticity were providing targets for mutation. The authors have improved the manuscript with a discussion of these issues and provided creative follow-up experiments that address the question empirically. Specifically, they investigate the mechanisms of aggregation under different conditions with biochemical experiments that demonstrate how different classes of molecules mediate inducible aggregation in the ancestor compared to the constitutive aggregation of derived strains. This result suggests that the mutations that stabilize the inducible phenotype occur in distinct pathways from those that mediate the induction of aggregation by salinity. It seems possible that more knowledge of the induction pathway could lead to experiments that provide examples of where that pathway is involved in stabilizing the phenotype. This result underscores the potential of this system to contribute to our knowledge of the role of plasticity in evolution.

(4) In my initial review, I had concerns about how the physical aggregation described in this study related to the appearance of new targets for evolution that are associated with a transition to multicellularity. The authors have addressed this concern with new experiments that provide a description of the life cycle of the aggregations produced in their study. Again, these experiments are creative and I find them convincing that that the phenomenon they describe is more significant than simple physical aggregation. As before, the changes to the manuscript further establish the potential value that this system can have for investigating the role of phenotypic plasticity in evolutionary change.

In summary, I commend the authors for the improvements they have made to their manuscript. These improvements both clarify their purpose and methods and provide additional evidence that their experimental system can be used to investigate the role of phenotypic plasticity in evolutionary change. Despite these improvements, the clarification that their selective conditions model the artificial selection used in agricultural breeding more closely than the natural selection that would have contributed to the appearance and persistence of multicellularity in nature undermines the value of this particular study.

RESPONSE TO REVIEWERS' COMMENTS

Reviewer #2 (Remarks to the Author):

Dear authors, first of all, I would like to apologize for taking an unreasonable amount of time to revise your manuscript a second time. I can see that you have put in a lot of work and made the paper much better. I really liked the clearer language and new experiments. Overall, I think this is an important contribution to the evolution of multicellularity literature, demonstrating the importance of previous plasticity. I think the paper is ready for publication.

Thank you very much!

Reviewer #3 (Remarks to the Author):

The authors have addressed most of my comments, and I am pleased that they intend to conduct a comprehensive study to investigate the genetic basis of filamentous group formation. I am happy with the publication of this study.

Thank you very much!

Reviewer #4 (Remarks to the Author):

The authors have submitted an improved manuscript and I remain excited about the high value of a tractable experimental system that can be used to investigate the role of phenotypic plasticity in evolutionary changes. In what follows, I will revisit the four comments I made on the author's first submission and evaluate their responses to my concerns.

Thank you!

(1) In my review of the original manuscript, I commented that the authors had not proven that phenotypic plasticity played a role in the physical aggregation that persisted as they withdrew salinity in the resting culture because the physical aggregations might have evolved in the absence of salinity-mediated induction. My understanding was that the authors were trying to demonstrate a process where phenotypic plasticity (ie, the clustering phenotype induced by high salinity) was scaffolding adaptation to inhabiting the air-liquid interface of the resting cultures. Based on that understanding, I interpreted the shaking cultures as a control designed to show the differences in evolution when the culture conditions did not provide a benefit (ie, increased availability of oxygen) at the air-liquid interface. I thank the authors for clarifying the process and intentions of their study.

The procedure is designed to exclude planktonically growing cells and allow only the transfer of physically aggregated biomass. The reason that the aggregated phenotype (inducible in the ancestor) is adaptive to the selected conditions is because the phenotypes most resistant to disaggregation under their washing procedure will be most likely to transfer to a new culture as salinity is withdrawn. The phenotype is adaptive under this transfer procedure in both the resting and shaken condition. Thus, we see that the genetic assimilation of the inducible phenotype proceeds under two different environments when the selective conditions are applied. The revised manuscript clarifies the purpose of their paper and their response explains why my original comment does not address their purpose.

Thank you. We are glad that our modifications have clarified this ambiguity.

However, these clarifications raise concerns about the validity of their procedure as it relates to the goals of their study. Rather than using phenotypic plasticity to provide variation to promote adaptation to conditions that favor multicellularity, the authors might be better characterized as demonstrating a technique for isolating mutations that make a formerly inducible phenotype into a constituent phenotype in the absence of induction. The authors characterize this process as "artificial selection." The question of whether the selective conditions imposed on a population in a laboratory experiment could be characterized as "natural" is somewhat philosophical, but for an experimental study to inform understanding of natural processes would depend on how well the laboratory selection process models natural selection outside the laboratory. The authors have not supported how a process that requires the judgement and conditional manipulation of the system by the experimenter (ie, identifying and isolating a portion of aggregated biomass to wash and transfer) is a valid model for the selective conditions that would have led to multicellularity in nature.

While this concern does not diminish the potential of this system to investigate the role of phenotypic plasticity during evolutionary change, it reduces the applicability of these particular experiments to understanding natural processes where there is not an intelligence to guide the trajectory of evolution. There are established conditions demonstrated to promote physical aggregation that model natural selection more closely, including increased oxygen availability at the air-liquid interface (Rainey and Travierno 1998 and subsequent work), using settling to select on particle size (Ratcliff et al 2012 and subsequent work), and predation (Boraas et al 1999 and subsequent work), and so demonstrating that variation introduced by phenotypic plasticity can contribute to the evolution of physical aggregation under any of these conditions would clearly represent an advance in the field. Introducing an unrelated artificial selection process is more ambiguous as to whether it is providing new insights into this evolutionary transition.

Two of the key goals of our study included the following:

1. To demonstrate that both Gram-negative and Gram-positive bacteria exhibit environmentally induced macroscopic cell clustering.
2. To show that phenotypically plastic cell clustering in ancestral genotypes can be rapidly assimilated to efficiently form multicellular clusters even in the absence of induction.

Upon fulfilling our first goal, we conducted artificial selection experiments with *E. coli* to address our second goal. As described in our results, unlike most other evolution experiments, here the phenotype of interest (macroscopic clustering) was already exhibited by the ancestor at the outset (induced by high salinity). We hypothesized that our artificial selection for macroscopic clustering under progressively reduced salinity would enrich mutations that can make such clustering genetically assimilated (and hence constitutive). This expectation mirrors the "genes as followers" view of phenotypic evolution. While our evolution experiments ended up addressing this goal, we appreciate that an experiment with pseudo-natural laboratory selection (e.g., for predation avoidance) would be an important next step. However, since the phenotype of interest is already available without new mutations at the outset of our evolution experiment, a conceptual extrapolation from our artificial selection to pseudo-natural selection favoring clustered macroscopic phenotypes should be quite straightforward. Specifically, a conceptual extrapolation where macroscopic clustering (which occurs at the outset without any mutations) is selectively favored in the face of predation by a eukaryote (instead of artificial selection by a human experimenter) is quite straightforward. Moreover, the ability of our clusters to undergo a life cycle in which a multicellular stage can spontaneously give rise to another multicellular stage via the propagation of small propagules should facilitate such an extrapolation to natural settings.

Taken together, while we appreciate that an empirical demonstration of genetic assimilation via pseudo-natural selection would be an important future direction, our demonstration of the genes-as-followers mode of evolution represents an important advance in the field of multicellularity.

(2) The authors are correct that my concerns about the differences between the aggregations present in the resting and shaken cultures do not apply to their study once the procedure and purpose is properly clarified. The changes I suggested in my first review are not needed and I appreciate their discussion of the differences between the observed outcomes in the resting and shaken cultures. These differences do make it plausible that the plastic aggregation might be acting as a scaffold for adaptation to the air-liquid interface in the resting cultures.

Thank you. We are glad that our modifications have clarified this point.

(3) In my review of the authors' first submission, I requested a discussion of how the pathways mediating a plastic response would function in evolutionary change. Specifically, I asked for a discussion on whether the plasticity was only providing variation that could be stabilized by any means or whether the pathways that mediated the plasticity were providing targets for mutation. The authors have improved the manuscript with a discussion of these issues and provided creative follow-up experiments that address the question empirically. Specifically, they investigate the mechanisms of aggregation under different conditions with biochemical experiments that demonstrate how different classes of molecules mediate inducible aggregation in the ancestor compared to the constitutive aggregation of derived strains. This result suggests that the mutations that stabilize the inducible phenotype occur in distinct pathways from those that mediate the induction of aggregation by salinity. It seems possible that more knowledge of the induction pathway could lead to experiments that provide examples of where that pathway is involved in stabilizing the phenotype. This result underscores the potential of this system to contribute to our knowledge of the role of plasticity in evolution.

Thank you. We are happy that our follow-up experiments have presented a convincing argument about the suitability of our experimental system to understand the role of phenotypic plasticity in evolution.

(4) In my initial review, I had concerns about how the physical aggregation described in this study related to the appearance of new targets for evolution that are associated with a transition to multicellularity. The authors have addressed this concern with new experiments that provide a description of the life cycle of the aggregations produced in their study. Again, these experiments are creative and I find them convincing that that the phenomenon they describe is more significant than simple physical aggregation. As before, the changes to the manuscript further establish the potential value that this system can have for investigating the role of phenotypic plasticity in evolutionary change.

Thank you. We are glad that our new experiments showing the time lapse of a life cycle has presented a strong case for the importance and novelty of our findings.

In summary, I commend the authors for the improvements they have made to their manuscript. These improvements both clarify their purpose and methods and provide additional evidence that their experimental system can be used to investigate the role of phenotypic plasticity in evolutionary change. Despite these improvements, the clarification that their selective conditions model the artificial selection used in agricultural breeding more

closely than the natural selection that would have contributed to the appearance and persistence of multicellularity in nature undermines the value of this particular study.

Thank you. While our study makes important contributions in the field of multicellularity by showing its 'genes-as-followers' mode of evolution, we appreciate that an important future direction includes demonstrating such genetic assimilation via pseudo-natural selection in the laboratory.